# Reading with deaf eyes: Automatic activation of speech-based phonology during word recognition is task dependent

Katherine Rowley[1]*, Eva Gutierrez-Sigut[2], Mairéad MacSweeney[1,3], Gabriella Vigliocco[4]

1 Deafness, Cognition and Language Research Centre, University College London, London, United Kingdom, 2 Department of Psychology, University of Essex, Wivenhoe Park, Colchester, United Kingdom, 3 Institute of Cognitive Neuroscience, University College London, London, United Kingdom, 4 Department of Experimental Psychology, University College London, London, United Kingdom

* kate.rowley@ucl.ac.uk

## Abstract

Literacy levels are highly variable within the deaf population and, compared to the general population, on average, reading levels are lower. As speech-based phonological coding is a known predictor of reading success in hearing individuals, much research has focussed on deaf readers' processing of speech-based phonological codes during word recognition and reading as a possible explanation for the widespread reading difficulties in the deaf population. Although results are mixed, there is recent growing evidence that deaf and hearing readers process speech-based phonological codes differently. Furthermore, some studies indicate that phonological ability may not be a strong correlate of literacy skills in deaf, adult readers. Here, we investigate orthographic, semantic, and phonological processing during single word reading in deaf (N = 20) and hearing (N = 20) adult readers, who were matched on reading level. Specifically, we tracked deaf and hearing readers' eye-movements using an adaptation of the visual world paradigm using written words and pictures. We found that deaf and hearing readers activate orthographic and semantic information following a similar time-course. However, there were differences in the way the groups processed phonology, with deaf readers making less use of phonological information. Crucially, as both groups were matched for reading level, reduced phonological processing did not appear to impact reading skill in deaf readers.

Literacy levels in the deaf population have consistently been reported as lower than in the hearing population in several countries including Spain [1,2], the USA [3,4], Canada [5] and the Netherlands [6]. In the United Kingdom, Conrad's [7] study showed that the mean reading-age of deaf 16-year-olds upon leaving school was, on average, equivalent to 9-year-old hearing children. Subsequent studies have shown

**Data availability statement:** All relevant data are within the paper and at the OSF (DOI 10.17605/OSF.IO/PNWKD).

**Funding:** KR: ESRC PhD Studentship, Award no: 1196158 ESRC (Economic and Social Research Council) https://www.ukri.org/councils/esrc/ The funders did not play any role.

**Competing interests:** The authors have declared that no competing interests exist.

little improvement [8–10]. However, many deaf individuals become excellent readers which suggests deafness itself is not a direct cause of poor literacy achievement [11–14]. Furthermore, recent evidence using highly sensitive techniques that allow monitoring of neural processing over time (e.g., event related potentials), suggests that the mechanisms involved in word recognition in deaf adults who are skilled readers might be different to the mechanisms used by hearing readers (for discussion see [13]). Gaining a better understanding of how deaf skilled readers recognise words can shed light on whether speech-based phonological processing is a necessary step for accurate word recognition in deaf adults.

## Phonological processing in deaf readers

Written language is built upon spoken language, with most writing systems being mapped, to some degree, onto the phonology of the spoken language that it represents. It has been argued that phonological awareness of speech is essential for successful reading development in hearing children [15–17]. Furthermore, reading deficits in hearing children are linked to poor phonological processing skills such as the inability to decode phonological information and to make connections between phonemes and graphemes [16,18,19].

Deaf readers' reduced access to the phonological structure of spoken language is often identified as a critical factor in explaining reading difficulties in the deaf population [4,5,14,20–22]. Some studies have suggested that deaf skilled readers can use phonology during word recognition [2,21,23,24]. However, there is also evidence that deaf adults who are skilled readers do not always use phonological information during reading, yet this does not impact reading skill [1,4,5,14,25]. In Table 1, we provide a summary of the main findings to-date. The table organises the findings in terms of whether deaf and hearing readers were matched for reading skills, the language used, the task, whether the task implicitly or explicitly recruits phonology and whether semantic processing is required. The implications of these differences across tasks are discussed in the next sections.

## Task requirements and differences in findings

Although there is some variability, in general, the pattern from previous studies suggests that if the task requires *explicit* phonological processing (e.g., lexical decision, rhyme judging or syllable counting), deaf adults are likely to show some evidence of recruiting phonological information [20,26], however, when the task requires implicit processing, phonological involvement is less clear [1].

Most studies testing phonological processing in deaf adults have used implicit tasks. The few that have used explicit tasks have been neuroimaging studies. For example, MacSweeney et al. [27] used a phonemic awareness task in which adult participants were asked to identify if two words had the same initial phoneme, e.g., gin-jug. Deaf and dyslexic participants showed some evidence of phonological awareness; however, they achieved much lower scores than their hearing, non-dyslexic counterparts. With similar results, Emmorey et al. [23] also carried out a phonemic awareness task, including judgement of word final phoneme. In both studies,

**Table 1. Results from previous studies examining phonological processing in deaf adults during reading related tasks.**

| Study | Matched Reading level between deaf and hearing readers? | Language | Experiment/Task | Implicit or Explicit? | Semantic processing required? | Evidence of phonological processing? |
|---|---|---|---|---|---|---|
| Belanger et al., (2013) | Yes (deaf skilled and less skilled readers) | English | Parafoveal preview benefits (invisible boundary paradigm) | Implicit | Yes | No |
| Belanger et al., (2012) Experiment 1 | No (deaf skilled and less skilled readers) | French | Masked Phonological Priming Lexical Decision Task | Implicit | No | No |
| Belanger et al., (2012) Experiment 2 | No (deaf skilled and less skilled readers) | French | Serial Recall Task | Implicit | No | No |
| Chamberlain (2002) Experiment 1 | No (deaf skilled and less skilled readers) | English | Spelling-to-sound correspondences | Implicit | No | No |
| Chamberlain (2002) Experiment 2 | No (deaf skilled and less skilled readers) | English | Lexical decision using pseudohomophones | Implicit | No | No |
| Cripps et al., (2005) | No (no reading measure used) | English | Masked Phonological Priming Lexical Decision Task | Implicit | No | No |
| Emmorey et al., (2013) | Yes (skilled) | English | Phonemic Awareness Task | Explicit | No | Yes |
| Hanson & Fowler (1987) | No (wide variety of reading skills) | English | Paired lexical decision task (rhyming/non-rhyming) | Implicit | No | Yes |
| Hanson et al., (1991) | No (wide variety of reading skills) | English | Semantic acceptability judgement task | Implicit | Yes | Yes |
| MacSweeney et al., (2009) | Yes (skilled) | English | Phonemic Awareness Task | Explicit | No | Yes |
| MacSweeney et al., (2013) | No (wide variety of reading skills) | English | Rhyme judgement task | Explicit | No | Yes |
| Gutierrez-Sigut et al., (2017; 2018) | Yes (wide variety of reading skills) | Spanish | Masked Phonological Priming Lexical Decision Task | Implicit | No | Yes |
| Fariña et al., (2017) | Yes (skilled) | Spanish | Single word presentation Lexical Decision Task | Implicit | No | No |
| Costello et al., (2021) | Yes (skilled) | Spanish | Phonological masked priming lexical decision task | Implicit | No | No |
| Costello et al., (2021) | Yes (skilled) | Spanish | Phonological masked priming semantic categorisation | Implicit | Yes | No |
| Sehyr et al., (2017) | Yes (skilled) | English | Serial recall | Implicit | No | Yes |
| Glezer et al., (2018) | Yes skilled | English | Phonological oddball detection (Amongst other tasks) | Explicit | Yes | Yes, but coarse-grained |

reading ability did not differ between deaf and hearing readers despite differences in phonological ability. In another neuroimaging study, also testing explicit phonological processing, Glezer et al. [28] found evidence of phonological processing in skilled deaf readers while they performed a phonological oddball detection task, i.e., participants had to press a button whenever they saw the letter string *xyz*. This was while making phonological, orthographic, and semantic decisions on letter strings presented to them in the scanner. The authors found that deaf readers showed less sensitivity to phonological manipulations compared to hearing readers.

In explicit tasks, different types of phonological awareness are recruited such as shallow (rhyme judgement) or deep (phonemic judgement). However, it seems that there is evidence of phonological awareness amongst deaf readers for both types, e.g., MacSweeney et al's [27] study (phonemic awareness) and in a later study (rhyme judgement) [26].

As already mentioned, when implicit tasks are used (such as masked lexical decision), the evidence for the use of phonology amongst deaf readers is less consistent than with explicit [5,13,29]. For example, using a lexical decision task with

words and pseudohomophones, Chamberlain [4] explored phonological processing in skilled and less skilled deaf readers as well as hearing readers matched in reading ability. Deaf skilled readers did not show any evidence of phonological processing. For the hearing and deaf less-skilled readers, there was a similar error pattern in rejecting pseudohomophones, suggesting both groups were influenced by phonology. One possible reason for this is that less skilled readers rely more on the phonological information available to them compared to skilled readers, which is a pattern often found in hearing young, beginner readers [19]. As orthographic knowledge increases amongst deaf readers through reading experience, sensitivity to visual and orthographic properties of words might play a more important role than phonological information in recognition [13,30] for recent discussions).

Also investigating implicit phonological processing, Belanger, Baum et al. [5] investigated both phonological and orthographic effects in a masked priming lexical decision task in French. There were four conditions; orthographically similar pseudohomophones (e.g., bore-BORD), orthographically dissimilar pseudohomophones (baur-BORD), orthographically dissimilar nonhomophonic nonwords (boin-BORD) and unrelated nonwords (clat-BORD). The results showed only an orthographic, but not a phonological, effect in both skilled and less skilled deaf readers.

EEG studies have also been used to explore implicit phonological processing in deaf readers. Using a "sandwich" masked priming paradigm and a lexical decision task, whilst collecting EEG data, Gutierrez-Sigut et al. [2] found both behavioural and electrophysiological evidence of early automatic phonological processing in deaf readers of Spanish with a wide variety of reading skills. Costello et al. [1], also using EEG, however, did not find phonological effects in skilled deaf readers of Spanish in a masked priming go-no-go task. The results of these studies show that even in languages with transparent orthographies, there is mixed evidence for the use of phonology.

In addition to whether the task requires explicit or only implicit access to phonology, the other dimension on which studies differ is whether the task requires access to semantic information. Most of the experimental tasks used require decisions that can be made based on familiarity of surface form (e.g., lexical decision), without obligatory semantic access. Table 1 shows that if the task recruits semantic information, deaf readers are less likely to recruit phonological information, however results are still mixed.

In a task that requires access to semantics, Hanson et al. [31] conducted a semantic acceptability judgement task using written tongue twister sentences and a memory interference task where participants recalled phonologically similar numbers. Both hearing and deaf participants made more errors with the tongue twister sentences and phonetically similar numbers, showing evidence of phonological processing. Although phonology was deemed important for reading, deaf participants had lower reading scores than the hearing participants, leaving the relationship between phonological and reading ability unclear for deaf readers.

In an eye-tracking study, Bélanger et al. [3] found that deaf readers did not make use of phonological information in a sentence-processing task (which requires access to meaning). In this study, the invisible boundary paradigm was used to measure the influence of orthographic, semantic, and phonological manipulations on eye-movements of deaf skilled and less skilled readers. Hearing readers, matched on reading level to the deaf skilled readers, were recruited as controls. Both deaf skilled and less skilled readers did not display any evidence of phonological processing, which lends further support to the hypothesis that deaf readers use phonological information less in tasks that require access to meaning, than those that do not [3].

These trends across different studies demonstrate that differences in the processing requirements of the task could explain the lack of consensus in the literature in relation to deaf adult readers' use of phonology. The present study uses an experimental paradigm that will enable us to test implicit phonological processing during single word reading in various tasks that require semantic activation.

## The relationship between phonological skills and reading ability in deaf readers

It has been argued that the relationship between phonological skills and reading ability is likely to be more complex in deaf than in hearing readers [10,13]. Even when there is evidence of deaf readers recruiting phonological information during

reading related tasks, deaf readers' performance on these tasks is generally worse than their hearing counterparts. For example, in the studies where deaf and hearing readers were matched on reading level, deaf readers achieved far lower accuracy scores on tasks of phonemic awareness compared to hearing readers [23,27]. This suggests that these deaf adults have established some level of spoken language phonological awareness, however not to the same level as hearing peers with the same reading skills.

Mayberry et al. [14] conducted a meta-analysis of 57 studies exploring the relationship between reading ability and phonological awareness in deaf children and adults. They reported that phonological coding and awareness skills explained only 11% of the variance in reading scores in the deaf participants. Mayberry et al. [14] concluded that 'PCA (phonological coding and awareness) skills are not the *sine qua non* of reading proficiency in the deaf population'. However, Mayer and Trezek [21] argued that phonological skills only made 12% contribution to reading variance for hearing readers, therefore cautioning that PCA is not dismissed as an important component of word recognition or reading skills in deaf readers.

Perhaps not surprisingly given the discussion above, whether the measure of use of phonology comes from an explicit or implicit task appears to be important to whether a relationship is observed. Gutierrez-Sigut et al. [2] found a significant positive correlation between reading ability (sentence reading and reading comprehension) and performance on a syllable counting task. Similar correlations between a syllable counting task and reading have been reported for adult deaf readers of Spanish [32] and English [33]. In contrast, when an implicit measure of phonological processing (masked priming) was used, Gutierrez-Sigut et al. [2] did not find a correlation between reading ability and behavioural or electrophysiological measures of automatic phonological processing in deaf readers, yet this was present in hearing skilled readers. Similarly, Sehyr et al. [34] used a serial recall task of phonologically similar or dissimilar words to measure implicit phonological processing. Deaf readers showed evidence of sensitivity to English phonology however this was not correlated with their reading ability. The pattern of results from these studies suggests a more complex relationship between reading ability and use of phonology in deaf than hearing readers, with phonological processing during implicit tasks being less related to reading ability in deaf than hearing readers.

The studies described so far have explored phonological processing in deaf readers of different abilities in order to determine its importance for literacy success. Findings from deaf less-skilled readers point to some degree of involvement of phonology. However, evidence from skilled readers is mixed and even where there is evidence of phonological processing, reading ability often does not correlate with those processes in adults [13,29,35]. The present study focuses on deaf skilled readers, who have reached the end state of literacy development, to investigate whether phonological processing occurs automatically during single word reading in the absence of an explicit task.

### Time-course of phonological activation

While the above behavioural studies have provided valuable insights into phonological processes in deaf readers they do not tap into the time-course activation of orthographic, semantic and phonological information which may be where deaf and hearing readers differ. Event-related potential studies using EEG have provided such information (e.g., [2]), however, the visual world eye-tracking paradigm provides another route for us to gain insights into the time-course of semantic, phonological, and orthographic processing during word reading. The visual world paradigm has been developed to investigate the time-course of phonological and semantic activation during spoken word recognition without relying on explicit tasks [36,37]. In visual world experiments with hearing participants, they are typically presented with an auditory target sentence (e.g., 'click on the fly') while simultaneously being showed an array of pictures including the target ('fly') and distractor pictures, which can be semantically related ('moth') and/or phonologically related ('sky') to the target, or unrelated to the target ('chair'). Participants' eye-movements from the onset of the auditory word are recorded and interpreted as providing information about the time-course activation of semantic and phonological information [37].

The visual world paradigm has also been used to explore written word recognition [38,39]. In particular, Kim et al. [40], was able to successfully identify differences in orthographic and phonological processing in adult readers of English. To our knowledge, this paradigm has not been used to explore written word recognition in deaf readers.

## The present study

The present study investigated orthographic, semantic, and phonological processing in severely/ profoundly deaf adults who use BSL (British Sign Language) as their primary language. To do this, we compared two groups of skilled readers, deaf and hearing, who were carefully matched for age, gender and reading level (on a pairwise basis) to ensure that any differences in semantic and phonological processing cannot be attributed to differences in reading ability. We used the visual world paradigm where written target words were presented in the centre of the screen along with four pictures, including target, semantic, phonological, and orthographic distractors. Eye-movements to the pictures were measured in order to explore the time-course of processing the different types of information.

Our use of the visual world paradigm allows us to assess if deaf readers recruit phonological information in tasks where semantic processing is required (word-to-picture matching). The two overarching goals of the work reported here are: [1] to investigate the time-course of orthographic, semantic and phonological activation during single word recognition during tasks that favour semantic processing (Experiments 1a and 2); [2] To investigate the role of phonology during single word recognition as a function of task demands (Experiments 1a and 1b). Below we present the results from three visual world experiments which use a visual world paradigm with different orthographic, semantic and phonological manipulations.

## Experiment 1a: Semantic and phonological processing in deaf and hearing readers during visual word recognition

Deaf and hearing readers were presented with written targets placed in the centre of a computer screen. Written targets were either words or nonwords. Nonwords were made up of both pseudohomophones and non-homophonic nonwords. Four pictures were presented at the same time as the word (see Fig 1). Previous work shows that there are differences in how deaf and hearing people manage visual information [13,29], particularly in the parafoveal regions, however, the pictures were presented more than 5–7 degrees from the centre minimising the chances of parafoveal benefits. These were phonologically related, semantically related, or unrelated to the words and pseudohomophones. If the target was a real word, the participant's task was to click on the target picture (i.e., the picture of the written target word). If the target was not a real word, they were instructed to simply wait for the next word (trial). Of primary interest was whether deaf and hearing readers differed in their eye-movements towards the pictures (target and distractors), and therefore the processing of phonological and semantic information, in the word and pseudohomophone conditions. It was important that target nonwords consisted of a mixture of pseudohomophonic (e.g., phly) and non-homophonic (e.g., adge) nonwords for comparison, as this helps to isolate phonological and orthographic processes in word recognition. Pseudohomophones

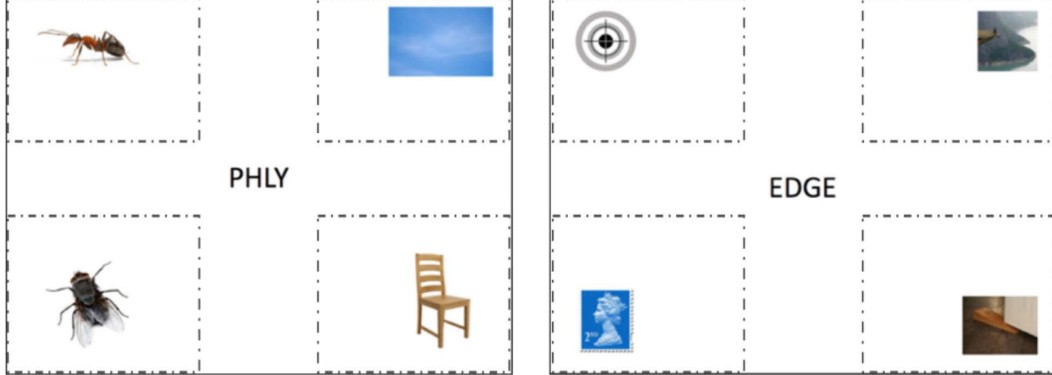

**Fig 1. The layout of the stimuli presented to participants with the four quadrants used for analysis.** Lines outlining the four quadrants were not shown.

measure the influence of phonology in lexical access, while non-homophonic nonwords measure the ability to decode unfamiliar letter patterns into a phonological word form but do not give lexical access to a known word. Any observed differences between these two conditions can be attributed to the use of phonology during lexical access. Based on existing literature, which have used priming and other paradigms, we hypothesise that deaf and hearing participants will exhibit similar behavioural strategies (e.g., eye movements and picture selection) when presented with word targets. However, we anticipate differences in their responses to pseudohomophone targets. As the non-homophonic nonwords are not orthographically or phonologically related to any of the pictures presented, we did not expect to observe any differences between deaf and hearing participants in this condition.

## Methods

**Participants.** 23 deaf adults who were severely/profoundly deaf and 25 hearing adults took part in this study, between 10th October 2014 and 31st January 2015. Hearing participants were native English speakers and did not know BSL. Participants from each group were matched on a one-to-one basis as closely as possible for age, gender, and reading level (see reading measures below). 1 deaf and 1 hearing participant were excluded because of no matched availability, following the set criteria. 2 deaf and 3 hearing participants achieved low scores (more than 2.5 standard deviations below the mean of their groups) on the reading test and were subsequently removed from the data analysis. Data from 1 hearing participant could not be used because of the corrupted data recording. Thus, 20 participants from each group (40 in total) were included in this study. Table 2 gives demographic information about both groups of participants. Table 3 provides relevant background information about the deaf participants, e.g., details on amplification aids. A power analysis was not conducted, as this study concerns a difficult to reach population, however our sample size aligns with previous studies [25,35]. Ethical approval for this study was granted by the Cognitive and Perceptual Brain Sciences (CPB) Ethics Committee at University College London (UCL) (ref: CPB/2012/002). All participants gave written informed consent prior to taking part.

*Reading measures.* Reading level was assessed using the Vernon-Warden Reading Test [41]. This is a timed reading test (10 minutes) in which participants choose the missing word in a sentence from the five options provided (max score = 42). A pairwise comparison of hearing/deaf readers' reading level showed no significant differences (t (40) = 1.229, p = 0.234). See Table 2 for the mean scores.

**Stimuli.** *Pseudohomophone targets:* We selected the pseudohomophones (e.g., phly) from previous studies that had found reliable phonological effects in hearing participants (i.e., Rastle & Brysbaert, 2006 and Twomey et al., 2011). We chose the items for which the target word was easily picturable and differed from the target word in at least one grapheme (e.g., fly/phly). Pictures were taken from clipart.com and were balanced on size and style. To ensure that the pictures we used elicited the expected labels, we asked 3 hearing native speakers of English (who did not take part in our main experiment) to name them. From 32 initially selected items we eliminated 4 four because their responses did not match the intended target. The final list was comprised of 28 pictures that elicited target English words, 28 matched

**Table 2.** *Participants' age, gender, and reading scores.*

| | Deaf | Hearing |
|---|---|---|
| Mean Age (years) | 31.5 | 30.75 |
| Reading age (years, months) | 18.04 (range, 15.04–23+) | 17.08 (range, 15.04–23+) |
| Mean Reading Raw Score* (max score 42) | 32.5 (SD = 5.26, range = 24–42) | 31.55 (SD = 4.55, range = 24–41) |
| Male | 8 | 8 |
| Female | 12 | 12 |

*A score of 32 is equivalent to a reading age of 18 years & 4 months [41].

**Table 3.** *Deaf participants' level of deafness (self-reported and unaided), amplification aids and sign language background (n = 20, note all deaf participants in this study were born deaf and use BSL as their main language on a daily basis. All participants reported that they were unable to have clear access to speech through a hearing aid).*

| Degree of Deafness | Profound – 15<br>Severe to profound – 3<br>Severe – 2 |
|---|---|
| Amplification aids | Hearing aids – 9<br>Cochlear Implants – 3<br>None used – 8 |
| Sign language background | Native – 11<br>*Near-native – 6<br>**Late learner – 3 |

*near-native signers in this study were exposed to sign language from age of 2 (i.e., attended signing deaf schools)

**Late learners used BSL as their main language for 10 + years at the point of testing and age of first exposure was at 11 years of age (secondary school).

pseudo-homophones and 28 control nonwords. See S1 Appendix for the full stimuli and distractor list. To ensure that visual saliency was evenly distributed across target and distractor images, three participants were asked to identify the image that most captured their attention or to indicate 'none' if no image stood out. On average, image selection ranged from 19.8% to 27.6% across target, semantic, phonological, and unrelated images, while no image was selected in 9.4% of cases. These results suggest that visual saliency was relatively balanced across target and distractor images.

*Word targets:* Besides the target words that matched the 28 pseudo homophones, we chose 28 additional picturable words (e.g., edge) from Balota et al., (2007) English Lexicon Project database. The additional words were added to account for visual/orthographic properties of pseudohomophones (e.g., 'phly' as a pseudohomophone of 'fly' is longer and visually different). Therefore, the additional words were identical to the pseudohomophones in length, orthographic neighbourhood size and closely matched on bigram frequency-by-position (t = 0.785, df (31), p = 0.439) (e.g., phly/edge). Both sets of words were similar in frequency (t = 0.978, df (31), p = 0.336) as measured using SUBLEX-UK [42].

*Non-word fillers:* A set of 28 non-homophonic nonwords (e.g., adge) were included as fillers to obtain the same number of trials in which there was a picture match (go trial) and in which there was no picture match (no-go trial).

Each word and pseudohomophone target were displayed to participants along with four pictures (see Fig 1). One picure corresponded to the target word/pseudohomophone. The other three pictures included a semantic distractor, a phonological distractor, and an unrelated picture (See Table 4).

Stimuli were counterbalanced so each participants saw both conditions with no repetition, but all stimuli were seen in all conditions across participants with the same pictures.

**Procedure.** Participants were tested individually in an acoustically adapted, dark room to reduce potential distractions. Participants' eye-movements were recorded using an *SR Research Eyelink 2 ©* system and *Experiment Builder Software (SR Research ©)*. Participants were placed 20–25 inches away from the monitor display. Prior to practice trials, calibration and validation of the eye-tracker was ensured. When needed, re-calibrations were carried out between trials. Each trial

**Table 4. Examples of items used.**

| Condition | Target stimuli | Target item | Semantic distractor | Phonological distractor | Unrelated item |
|---|---|---|---|---|---|
| Word | fly | fly | ant | sky | chair |
| Pseudohomophone | phly | fly | ant | sky | chair |
| Word | edge | edge | target | wedge | stamp |
| Non-homophonic nonword | adge | edge | target | wedge | stamp |

began with a fixation cross placed in the centre of the screen, as soon as the participant's gaze was on the fixation, the letter string was shown in the centre, surrounded by four pictures. After the first 500 ms, the letter string disappeared with the pictures remaining on-screen until participants responded or until 4.5 seconds was elapsed (See Fig 2).

Participants were instructed to keep their hand on the mouse to click on the corresponding picture quickly if the stimulus was a word or to wait for the next trial if a nonword. There were 8 practice trials (4 words and 4 nonwords/pseudo-homophones) and 56 experimental trials. We chose to use different instructions and tasks for words/nonwords because of the difficulty in choosing a task that could be equally applied across conditions. Trial order was randomised to reduce the potential impact of participants using different anticipatory strategies for the word or non-word tasks. Moreover, direct statistical comparisons across word and nonword conditions are not conducted.

**Data analyses.** Data from each participant's right eye were analysed. The proportion of fixation samples (how many fixations on the picture of interest in comparison to the other three pictures, in each time bin of 100 ms) for each quadrant in a single experimental trial (see Fig 2) were coded for analysis in 100 ms bins starting from target word-onset to 2000 ms. Note that only analyses from 400 ms post-target onset are presented here as there were not enough looks at the quadrants before 400ms. This is likely due to the use of a visual target presented for 500 ms. Therefore, fixations up to 400 ms were to the target for both groups. Based on the average response times of 2300 ms, analyses ended at 2000 ms to avoid capturing any post-recognition strategic processing and including trials of different lengths in the analyses. Separate analyses were carried out for word (go) and pseudohomophone (no go) items. The dependent variable in each

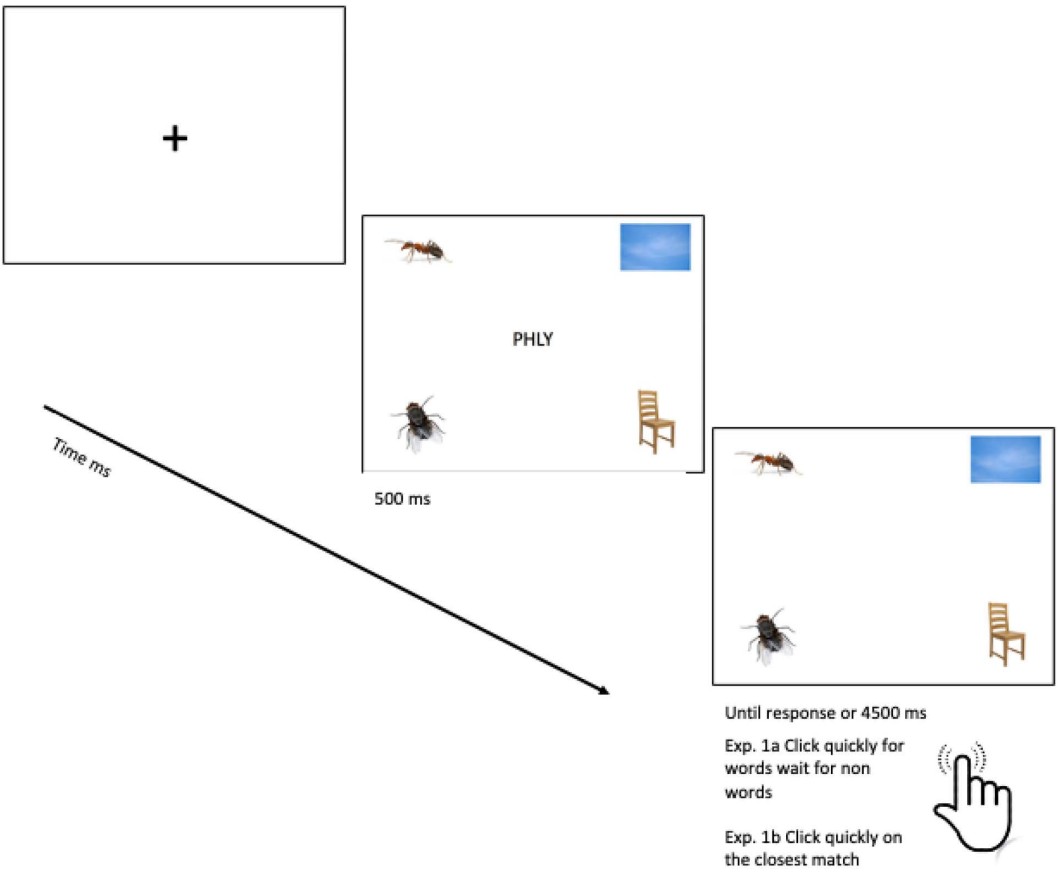

**Fig 2. Timeline of a single experimental trial.**

analysis was proportion of fixations to each of the four quadrants of the screen, corresponding to location of presentation of different picture types (see Fig 1).

We used the Growth Curve Analysis [43] in R version 3.3.3 [44] and the lme4 package v.1.1–12 and lmerTest package [45] to identify whether the pattern of looks to each condition (target, semantic and phonologically related) was different to the unrelated over the analysis window. More importantly for this study, possible differences between the groups in the proportion of looks over the analysis period were assessed by introducing the interaction with group. We modelled the proportion of fixations for each condition (target, phonologically, or semantically related) using orthogonal polynomials (first, second, third and fourth order polynomials). The model included the condition (target, semantically or phonologically related distractors) as fixed effects, adopted a maximal random effect structure, and included all correlations. If the maximal random effects and correlations model did not converge, correlations and then random effects were removed one at the time until the model converged [46].

In the Growth Curve Analyses, the polynomials map to the shape of the function (e.g., linear, quadratic, etc.) [43] rather than to specific time points. Significant results in the GCA would indicate that there are differences over time but does not specify when. Therefore, we followed up with the time bins analysis in order to understand the temporal dynamics better than just observing the differences in the plots.

To do this we used Generalised Linear Mixed Models (GLMMs), exploiting lme4 v.1.1−26 and lmerTest v.3.1−3 packages [45] to explore in detail the time course of looks to the different conditions (vs. unrelated) in each of the groups separately. As above, we chose the most complex model that converged in terms of random effects intercepts and slopes. The Bonferroni correction was used for multiple comparisons in all analyses.

## Results

### Word condition

Word trials with incorrect responses (in which the wrong target items were selected) were not included in the data analysis, making up 5% of the total data. As seen in Fig 3, fixations on target items were higher than on any other distractors for both the deaf and hearing groups. Fixations on semantic distractors were higher compared to phonological distractors and unrelated items.

*Fixations to target items.* In the growth curve analyses there was a significant effect of target on the first, second and fourth polynomials (ps < 0.01; see Table 5) indicating a different pattern of fixations to the target and unrelated conditions over the analysed time window. The proportion of looks to the target was higher than to the unrelated over the time window. Importantly, neither the effect of group nor any of the interactions between group and the time polynomials were significant (all ps > 0.05), indicating that the temporal profile of the target effect did not differ between deaf and hearing readers. The bin-by-bin GLMM analyses revealed that both deaf and hearing participants had fixations on target items significantly more than unrelated items in most time-windows: with Deaf showing significant differences after Bonferroni correction between 600 and 2000 ms and hearing between 700–2000 ms.

*Fixations to semantically-related distractors.* There was a significant effect of semantics on the first to fourth polynomials (all ps < 0.01; see Table 5). There was no significant effect of group. Importantly, there was no effect of group nor any significant interactions between group and time polynomials, which indicates that the temporal profile of the semantic effect did not differ for deaf and hearing readers. The bin-by-bin GLMMs analyses showed that after Bonferroni correction deaf participants had more fixations on the semantic distractor than the unrelated between 600 and 1200ms, while hearing participants had more fixations on the semantic distractor than the unrelated between a similar time window (700–1200 ms), and slightly later during the 1300–1400 ms time bin.

*Fixations to phonologically-related distractors.* There was a significant effect of phonology on the first to forth polynomial (all ps < 0.01; see Table 5) as well as an interaction between group and the third time polynomial (p < .05), which indicates that the temporal profile of the phonological effect differed for deaf and hearing readers. However, the bin-by-bin GLMMs analyses did not show any significant effect that survived the multiple comparisons correction in either group.

**Fig 3. Results from the word and pseudohomophone conditions.** Graphs display the proportion of fixation samples between 0 ms and 2000 ms for deaf and hearing readers in each condition. Here the labels "pseudotarget", "pseudosemantic" and "pseudophonological" refer to the relationship of the picture to the word from which the pseudohomophone is derived, e.g., kote (pseudotarget – from coat), shirt (semantic), and boat (phonological). Summary tables below the graphs show significant differences in the bin-by-bin analyses between the unrelated condition and each of the remaining distractor conditions: target (T), semantically related (S), and phonologically related (R) for hearing and deaf participants separately for word and pseudowords.

## Pseudohomophone condition

Fig 3 shows with fixations, hearing readers were mostly drawn to pseudotargets in comparison to the other distractor items. However, the pattern is less clear for deaf readers.

*Fixations to (pseudo)target items* – There was a significant effect of pseudotarget item on the second polynomial (Estimate = −0.52, SE = 0.14, p < 0.01). There was no effect of group nor interaction between group and the time polynomials (See Table 6). The detailed bin-by-bin GLMM analyses showed that after Bonferroni corrections for multiple comparisons hearing participants fixated on pseudotarget pictures significantly more than on pseudounrelated pictures between 1000 and 1400 ms. No differences were found for deaf readers at any time bin.

*Fixation to (pseudo)semantically-related distractors.* There was a significant effect of pseudosemantics on the second polynomial (Estimate = −0.27, SE = 0.09, p < 0.01). There was no significant effect of group (Estimate = 0.02, SE = 0.01, p = 0.06). There was an interaction between group and one of the time polynomials (Estimate = −0.64, SE = 0.17, p < 0.01),

Table 5. Parameter estimates for each time polynomial, group and interactions for each time polynomial and group for target items, semantic and phonological distractors in the word condition.

| Target Items | | | | | Semantic Distractors | | | | Phonological Distractors | | | |
|---|---|---|---|---|---|---|---|---|---|---|---|---|
| **Fixed Effects** | Estimate | SE | T value | P value | **Estimate** | SE | T value | P value | **Estimate** | SE | T value | P value |
| **Intercept** | 0.40 | 0.01 | 32.50 | <0.01 | 0.06 | 0.00 | 18.92 | <0.01 | 6.53 | 3.43 | 19.04 | <.01 |
| **Time polynomial 1** | 4.38 | 0.20 | 21.74 | **<0.01** | −0.20 | 0.08 | −2.60 | **0.01** | −2.01 | 7.61 | −2.64 | **<0.01** |
| **Time polynomial 2** | −1.77 | 0.16 | −10.76 | **<0.01** | −0.36 | 0.06 | −5.90 | **<.01** | −3.61 | 4.55 | −7.93 | **<.01** |
| **Time polynomial 3** | 0.19 | 0.12 | 1.56 | 0.13 | 0.52 | 0.06 | 8.75 | **<.01** | 5.21 | 4.55 | 11.46 | **<.01** |
| **Time polynomial 4** | 0.26 | 0.10 | 2.67 | **0.01** | −0.16 | 0.06 | −2.52 | **0.02** | −1.57 | 4.55 | −3.44 | **<.01** |
| **Group** | −0.05 | 0.02 | −1.83 | 0.07 | −0.00 | 0.01 | −0.03 | 0.98 | −1.84 | 6.85 | −0.03 | 0.98 |
| **Time polynomial 1 x Group** | −0.12 | 0.40 | −0.29 | 0.77 | 0.11 | 0.15 | 0.71 | 0.48 | 1.10 | 1.52 | 0.72 | 0.48 |
| **Time polynomial 2 x Group** | 0.35 | 0.33 | 1.07 | 0.29 | −0.13 | 0.12 | −1.04 | 0.30 | −1.27 | 9.10 | −1.40 | 0.16 |
| **Time polynomial 3 x Group** | 0.06 | 0.24 | 0.27 | 0.79 | −0.19 | 0.12 | −1.55 | 0.13 | −1.85 | 9.10 | −2.04 | **0.04** |
| **Time polynomial 4 x Group** | 0.25 | 0.20 | 1.25 | 0.21 | −0.09 | 0.12 | −0.76 | 0.45 | −9.39 | 9.10 | −1.03 | 0.30 |

Table 6. Parameter estimates for each time polynomial, group and interactions for each time polynomial and group for pseudotarget items, pseudosemantic and pseudophonological distractors in the pseudohomophone condition.

| Pseudotarget Items | | | | | Pseudo -semantic Distractors | | | | Pseudo -Phonological Distractors | | | |
|---|---|---|---|---|---|---|---|---|---|---|---|---|
| **Fixed Effects** | Estimate | SE | T value | P value | **Estimate** | SE | T value | P value | Estimate | SE | T value | P value |
| **Intercept** | 0.03 | 0.01 | 5.09 | <0.001 | 0.02 | 0.00 | 3.31 | <0.001 | 0.01 | 0.00 | 2.01 | 0.05 |
| **Time polynomial 1** | 0.09 | 0.11 | 0.80 | 0.43 | 0.13 | 0.11 | 1.21 | 0.23 | 0.23 | 0.12 | 1.98 | 0.06 |
| **Time polynomial 2** | −0.52 | 0.14 | −3.65 | **<0.001** | −0.27 | 0.09 | −3.17 | **0.001** | −0.40 | 0.08 | −1.80 | **0.07** |
| **Group** | 0.02 | 0.01 | 1.56 | 0.13 | 0.02 | 0.01 | 1.92 | 0.06 | −0.00 | 0.01 | −0.24 | 0.81 |
| **Time polynomial 1 x Group** | −0.17 | 0.22 | −0.73 | 0.46 | 0.21 | 0.22 | 0.96 | 0.34 | 0.15 | 0.24 | 0.61 | 0.54 |
| **Time polynomial 2 x Group** | −0.47 | 0.29 | −1.64 | 0.11 | −064 | 0.17 | −3.76 | **<0.001** | −0.09 | 0.16 | −0.56 | 0.58 |

which means that the temporal trajectory of the pseudosemantic effect differed for deaf and hearing readers (See Table 6). The bin-by-bin GLMM analyses did not show any significant effect after Bonferroni correction for any of the groups. *Fixation to (pseudo)phonological distractors.* In the growth curve analyses there was no effect of pseudophonological distractors for any of the time polynomials. There was no effect of group, nor interactions between the time polynomials and group (see Table 6). The bin-by-bin GLMM analyses showed no significant effects after Bonferroni correction in any time-window.

## Discussion – Experiment 1a

Eye-movements from two groups of carefully matched deaf and hearing adults were recorded to investigate semantic and phonological activation during the single word reading using a modification of the visual world paradigm. The aim was to determine whether deaf skilled readers automatically activate phonological, in addition to semantic information when the task focuses on meaning, and to identify the time-course of this activation for each group. In one condition, we used real words (e.g., *fly*) as stimuli, and in the other, we used pseudohomophones (e.g., *phly*).

When the target was a real word (*fly*), there was very little difference in target fixations between deaf and hearing participants to the images representing the target words (a fly). Both groups had fixations on target items significantly more than unrelated items (e.g., picture of a *chair*) starting at a similar time (600 ms post word onset for the deaf and 700ms for

the hearing group) until the end of the analysis window. Both groups also had higher proportion of fixations to the semantic (e.g., picture of an *ant*) than to the unrelated distractor (*chair*), however there was a subtle difference between the groups. Specifically, the hearing group kept looking at the semantic more than the unrelated distractor for a longer time than the deaf group. Regarding the phonologically related distractors (e.g., picture of the *sky*), the growth curve analyses showed that there were more fixations to the phonologically related distractor (*sky*) than the unrelated (*chair*) distractor. Interestingly, the growth curve analysis suggested a different pattern of fixations for both groups across the period of analysis. Although visual inspection of the graphs might suggest that the hearing participant's effect might be earlier and more extended than the effect for the deaf group, the proportion of looks to both conditions was low, and no effect survived correction for multiple comparisons in the time-bin analyses.

When the target word was a pseudohomophone (*phly*), and hence the participants were instructed to not click on any of the images, readers had a larger proportion of fixations to the pseudotarget (*fly*) than to the unrelated (*chair*). The growth curve analysis suggested that the effect was in the same direction and developed following a similar curve over the whole analysis period in both groups. Importantly, the time-bin analyses revealed that the effect was robust for hearing readers between 100 and 1400 ms after target onset, indicating that even in the absence of a task, hearing readers could process the pseudohomophone to some extent and hence be drawn to the target picture. However, this was not the case for deaf readers who did not show significant differences in fixations between the target and unrelated pictures at any time window. This finding is in line with other studies that did not find an effect of phonology in deaf skilled readers during single word recognition [5,25].

However, there is a possibility that the go/no go task might have had different consequences for each group. For example, inducing the use of different strategies in both groups in the pseudohomophone condition where a quick orthographic analysis could be enough for deaf readers to decide that they did not need to look at any of the distractors because no response was required. This response is congruent with papers that show deaf readers are more sensitive to the visual/orthographic aspects of written words, which has been shown to influence electrophysical components linked to both early processing and lexical access [30]. To address this concern a follow-up experiment was carried out in which only pseudohomophones were shown to the two groups and a response was always required.

## Experiment 1b: Exploring semantic and phonological processing in deaf readers using the visual world paradigm in a forced choice task

In experiment 1b deaf and hearing participants were presented *only* with written pseudohomophones. As in experiment 1a, their task was to select the picture that they felt was the closest match to the nonword presented. Therefore, participants were forced to select and respond to all stimuli. This may lead participants to adopt a different strategy to that observed in Experiment 1a. As previous studies suggest that deaf readers can process phonological information, albeit to a lesser extent than hearing readers [28], we hypothesise that deaf participants will demonstrate phonological processing when presented with a pseudohomophone by selecting the corresponding picture. However, we also predict that deaf readers will show fewer fixations on the target picture than hearing readers, indicating reduced phonological processing.

### Method

**Participants.** See participant section in Experiment 1a outlining the participant criteria for this experiment. Twenty-one deaf adults with 12 being from Experiment 1a and 24 hearing speakers of English took part in this study between 1st September and 30th November 2016. 1 deaf participant was excluded because of their moderate deafness, including her low score achieved on the reading test (more than 2.5 SDs below the group mean). 2 hearing participants were excluded because of not being matched to deaf readers on the criteria. Data from 2 hearing participants could not be used in the analyses because of data recording corruption. This resulted in 20 participants from each group (40 in total) being included. Table 7 provides participant demographic information. Table 8 provides information about deaf participants' deafness level, use of amplification aids and sign language background.

 

**Table 7.  *Participants' age, gender, and reading level.***

|  | Deaf | Hearing |
|---|---|---|
| Average age | 33.85 | 34.05 |
| Reading score (raw score, max 42) | 32.35 (sd, 4.94; range, 23–42) | 31.60 (sd, 4.19; range, 24–42) |
| Reading age (years, months) | 18.04 (range, 15.0–23+) | 17.08 (range, 15.04–23+) |
| Male | 8 | 8 |
| Female | 12 | 12 |

**Table 8.  *Deaf participants' deafness, amplification aids and language background.***

| Deafness | Profound – 17<br>Severe to profound – 3 |
|---|---|
| Amplification aids | Hearing aids – 7<br>Cochlear Implants – 4<br>None used – 9 |
| Sign language background | Native – 11<br>*Near native – 6<br>**Late learner (– 3 |

*near-native signers in this study were exposed to sign language from age of 2 (i.e., attended signing deaf schools).

**late learners used BSL as main language for more than 10 years and age of first exposure was at 11 years of age (secondary school).

**Background measures.**  See background measures section in Experiment 1a for description of background measures used.

*Reading measures.* A pairwise comparison of the reading level of deaf and hearing readers showed no significant differences *(t (40) = 1.543, p = 0.139)*. See Table 7 for scores.

**Stimuli.**  The 28 pseudohomophones (e.g., phly) used in Experiment 1a were used again for this experiment, however in this instance there was no counterbalancing thus all participants saw all 28 pseudohomophones (S1 Appendix). There were no non-homophonic nonwords (e.g., adge) used in this experiment.

**Procedure.**  See procedure outlined in Experiment 1a for description of experiment set up (Fig 2). There were 4 practice trials and 28 experimental trials. Participants were instructed to click on the picture that they felt was the closest match to the nonword presented to them. Participants were not instructed about the nonwords being pseudohomophones.

**Data analyses.**  Analyses were carried out as in Experiment 1a, including all time bins between 0 and 2500ms. The upper cutoff was selected based on behavioural responses being slower for the pseudowords than the words (mean of 2700ms). Trials with incorrect responses (21% of the total data in which the wrong target items were selected) were not included in data analysis.

**Results.**  *Fixations to pseudotarget items*. There was a significant effect on first, second and third polynomials (all ps < .001), but no effect of group or interaction between the polynomials and group (See Table 9 and Fig 4), which indicates that there was higher proportion of fixations on pseudotarget items compared to unrelated items for both groups. The bin-by-bin GLMM analyses also showed that both groups had higher fixations on pseudotarget items compared to unrelated items in most time-windows: deaf 0–100 ms and 200–2500 ms and hearing 100–2500 ms.

*Fixation to pseudosemantic distractors*. Fig 4 shows that there was higher proportion of fixations on pseudosemantic distractors compared to unrelated items for both groups. There was a significant effect of pseudosemantic distractors on the first and second polynomials (both ps > .001). When group was included as a factor, there was no significant effect of group (Estimate = .01, SE = .01, p = 0.43). There were no interactions between group and any of the time polynomials (See

**Table 9. Parameter estimates for each time polynomial, group and interactions for each time polynomial and group for pseudotarget items, pseudosemantic and pseudophonological distractors in the pseudohomophone condition.**

| Pseudotarget Items | | | | | Pseudo – semantic Distractors | | | | Pseudo – phonological Distractors | | | |
|---|---|---|---|---|---|---|---|---|---|---|---|---|
| **Fixed Effects** | Estimate | SE | T value | P value | **Estimate** | SE | T value | P value | **Estimate** | SE | T value | P value |
| **Intercept** | 0.47 | 0.02 | 23.86 | <0.001 | 0.04 | 0.01 | 5.70 | <0.001 | 0.02 | 0.01 | 2.67 | 0.01 |
| **Time polynomial 1** | 3.66 | 0.61 | 5.58 | **<0.001** | −0.86 | 0.23 | −3.74 | **<0.001** | 0.12 | 0.35 | 0.35 | 0.72 |
| **Time polynomial 2** | −2.68 | 0.39 | −6.94 | **<0.001** | −0.59 | 0.18 | −3.23 | **<0.001** | −0.06 | 0.32 | −0.19 | 0.85 |
| **Time polynomial 3** | 0.99 | 0.17 | 5.81 | **<0.001** | | | | | 0.41 | 0.14 | 2.94 | **<0.001** |
| **Time polynomial 4** | | | | | | | | | 0.32 | 0.01 | 1.46 | **0.02** |
| **Group** | 0.04 | 0.04 | 0.09 | 0.35 | 0.01 | 0.01 | 0.79 | 0.43 | 0.02 | 0.01 | 1.46 | 0.15 |
| **Time polynomial 1 x Group** | −0.71 | 1.22 | −0.58 | 0.56 | −0.01 | 0.46 | −0.02 | 0.99 | 0.90 | 0.71 | 1.27 | 0.21 |
| **Time polynomial 2 x Group** | −1.23 | 0.77 | −1.59 | 0.12 | −0.44 | 0.37 | −1.19 | 0.24 | 0.38 | 0.64 | 0.61 | 0.55 |
| **Time polynomial 3 x Group** | 0.10 | 0.34 | 0.28 | 0.78 | | | | | 0.42 | 0.28 | 1.49 | 0.14 |
| **Time polynomial 4 x Group** | | | | | | | | | | | | |

## Experiment 1b

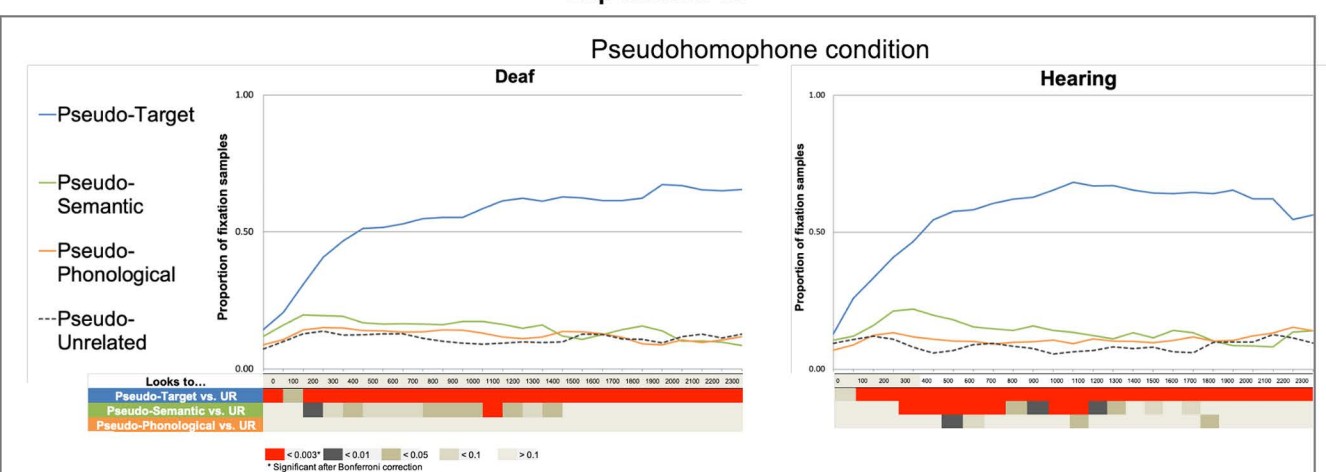

**Fig 4. Proportion of fixation samples between 0 ms and 2500 ms for deaf readers and hearing readers for Experiment 1a.** Here the labels "pseudotarget", "pseudosemantic" and "pseudophonological" refer to the relationship of the picture to the word from which the pseudohomophone is derived. (E.g., kote (from coat), shirt (semantic), and boat (phonological). Summary tables below the graphs show significant differences in the bin-by-bin analyses between the unrelated condition and each of the remaining distractor conditions: target (T), semantically related (S), and phonologically related (R) for hearing participants and deaf participants separately for word and pseudowords.

Table 9). The bin-by-bin GLMM analyses showed that although fixations to pseudosemantic distractors go in the same direction (i.e., higher than for unrelated) in both groups, for the hearing participants fixations on pseudosemantic pictures were significantly more than unrelated items in many time-windows starting at 300 ms (300–800 ms, 1000–1200 ms). For the deaf participants, the difference between pseudosemantic and unrelated items was only significant at a later time-window (1100–1200 ms).

*Fixations to pseudophonological distractors.* There was a significant effect of pseudophonological distractors on third and fourth polynomials (both ps < .05). There was no significant effect of group (Estimate = .02, SE = .01, p = 0.15) nor

interactions between group and time polynomials. The bin-by-bin GLMM analyses showed that there were no robust significant differences in fixations on pseudophonological distractors and unrelated items across all time-windows for either group (see Fig 4).

## Discussion – Experiment 1b

This study investigated whether deaf and hearing readers differed in eye movements when forced to make a decision based on phonological information. Although participants were not informed of the phonological manipulation, they were presented only with written pseudohomophones (*phly*) and they were asked to click on the picture that they felt was the best match to the nonword shown. The error rate in this study was high (21%). It is likely that this is due to the nature of the task. Participants were required to select the picture that best matched the presented pseudohomophone, without encountering any real words during the experiment. Since pseudohomophones are not actual words and the task demanded quick responses, this likely contributed to the high error rate.

Both deaf and hearing readers had significantly more fixations on pseudotarget (e.g., picture of a fly) pictures than on unrelated items (e.g., picture of a chair) across most time-windows from as early as 100ms from stimulus onset. The pattern of fixations as well as the strength of the effect was similar for deaf and hearing readers, showing that both groups were able to process the pseudohomophone when the task required an explicit response, and the phonological information could help to successfully identify the pseudotarget item (picture of a fly). The comparison between pseudosemantic (picture of an ant) distractors and pseudounrelated (picture of a chair) items showed that both groups looked at the pseudosemantic distractor more than the unrelated items. The time bin analyses indicated that while the overall behaviour (i.e., more looks to pseudosemantic distractors) was similar between deaf and hearing people, there can be subtle differences in processing between the groups. Specifically, the hearing readers had fixations on pseudosemantic distractors significantly more than deaf readers during several time-windows starting at 300 ms but the difference was not robust enough to survive multiple comparisons at most time windows for the deaf readers.

Together, the results from Experiment 1a and 1b, show that both deaf and hearing readers' fixations on target and distractor items in each condition was extremely similar. That is, the target word strongly drove the looks towards the target picture, then towards the semantically and phonologically related pictures more than to the unrelated. There were some indications that the semantic and phonological effects elicited by the target words could be stronger and/or longer lasting for the hearing than the deaf readers.

We did not inform participants that they would be viewing pseudohomophones. However, it is likely that they became aware of them, possibly leading to more overt phonological coding. This is noteworthy, as it demonstrates that deaf readers can access phonology and will utilise it when required by the task. HHowever, as demonstrated in Experiment 1a, phonological processing does not appear to be automatic. These findings align with our conclusion: while deaf readers can and do use phonology when necessary for semantic access, it is not their most efficient processing pathway.

When the target was a pseudohomophone (phly), there were some small but noteworthy effects. Experiment 1a showed that hearing readers clearly had a higher proportion of fixations on pseudotarget items (picture of a fly) than unrelated items, likely reflecting automatic activation of phonological features of the target word. Deaf readers showed a similar pattern of looks across the analyses window, hence there was not an interaction in the growth curve analyses. However, when deaf readers were analysed separately in the time-bin analyses, the difference between proportion of fixations to the pseudotarget and the unrelated distractors was not strong enough to be significant at any time window. The later finding changed when the experimental paradigm required a response in Experiment 1b. When required to click on the closest match to the pseudohomophone word, both groups of readers clearly had a larger proportion of fixations on the pseudotarget item (fly picture). This phonological effect was strong for both groups at most time windows, showing that when the use of phonological information is relevant for the task at hand, deaf readers access

phonological information from written characters early during processing. Potential subtle differences between hearing and deaf readers lie in the extent to which the information provided by the pseudohomophone target (phly) can strongly and quickly activate the semantic network of the corresponding word (fly). While this semantic effect was clear at many time windows for hearing readers it might be less robust for deaf readers upon viewing pseudohomophones. This finding is in line with some other studies that show deaf readers can and do process phonology but less so compared to hearing readers [28].

These results are important, as they show that despite differences in language experience, processing of words is very similar between skilled deaf and hearing readers when the phonological information from words is required to successfully complete a task. This finding is consistent with previous studies that have shown a phonological effect using an explicit task [23,28] indicating that deaf readers can and do process phonological information when required. Our findings also show that deaf readers when phonology is not useful to perform the task, task as in Experiment 1a, deaf readers seem to use it less than hearing readers. The latter finding can explain why some studies using implicit tasks and measuring automatic processing have not found phonological effects in deaf skilled readers [5,25].

## Experiment 2: Comparing orthographic and phonological processing in deaf and hearing readers using the visual-world paradigm

The previous experiments demonstrate similarities in the semantic and phonological processing of deaf and hearing skilled readers especially when targets were real words. This shows that the connections between orthography and semantics are robust for both groups. However, there were some subtle differences when pseudohomophones were presented that suggest that for deaf readers, the connections between phonological and semantic processing are reduced compared to hearing readers. This is not surprising, as deaf readers have reduced access to the spoken language phonology.

Previous studies have reported that deaf adult readers rely more on orthographic than phonological information and skilled deaf readers have a stronger connection between orthography and the lexical-semantic level of processing than typical hearing readers [3,5,47]. While in Experiment 1a, we found that connections between orthography and semantics are robust for both groups, we did not include orthographically similar distractors thus cannot draw strong conclusions. Here, we compare phonological and orthographic processing using real words along with homophonic (e.g., knight/night) and orthographically similar distractors (e.g., boat/boot).

If deaf readers activate orthographic information more efficiently than hearing readers, we may find differences between the groups in their fixations on orthographic distractors. If hearing readers activate phonology more strongly than deaf readers, as suggested by the findings in Experiments 1a and 1b, we should find differences in fixations on homophonic distractors between the two groups.

### Method

**Participants, Background measures.** The same participants who took part in Experiment 1a, also took part in Experiment 2 (see Tables 7–8).

**Stimuli.** There were two conditions: homophone and orthographic. For each condition, homophonic and orthographic similar pairs were created. Prior to commencing the study, we asked 3 additional participants (who did not take part in the main experiment) to name them to ensure that the pictures we used elicited the labels we expected. From the initial 19 homophonic pairs, 3 pairs were removed as responses did not match the intended target. From our initial 22 orthographically similar pairs, 2 pairs were removed as responses did not match the intended target (see S2 Appendix for the complete list of stimuli).

*Homophones.* Participants were presented with a target word (e.g., night) along with four pictures (see Fig 5), one of which was a homophonic distractor (e.g., knight). In this condition, there was also a distractor that was semantically

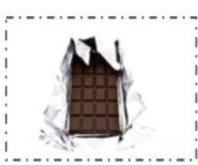 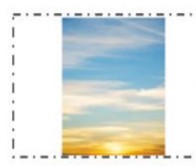 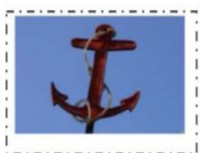 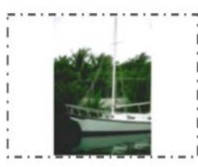

NIGHT                          BOAT

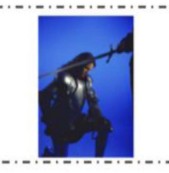 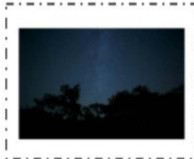 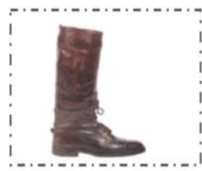 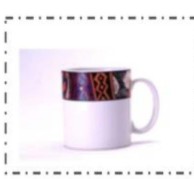

Homophone condition                    Orthographic condition

**Fig 5. Layout of the stimuli presented to participants in the homophone condition (Description of stimuli from top-left to bottom-right; chocolate (unrelated), day (semantic), knight (homophonic) and night (target) and in the orthographic condition (Description of stimuli from top-left to bottom-right; anchor (semantic), boat (target), boot (orthographically similar) and mug (unrelated).**

related to the target item (e.g., day) and an unrelated item (e.g., chocolate). In total, there were 16 sets of homophones. Since there was a significant difference in word frequency of the target and the homophone pairs $(t (15) = 2.321, p = .035)$, we created two lists: in the first list, participants saw 'night' as the target and 'knight' as the homophone and in the second list, participants saw 'knight' as the target and 'night' as the homophone. The use of these counterbalancing lists alleviates issues related to differences in frequency between homophonic pairs. Semantic distractors were selected to match the target that each participant saw. In the example above, the semantic distractor was changed from 'day' to 'castle' whereas the unrelated item remained as 'chocolate'. Participants were only shown one list.

*Orthographically similar items.* As with the homophones, participants were presented with a target written word on the screen surrounded with four pictures – an orthographically similar item, a semantically related item, and an unrelated item (See Fig 5). There were 20 sets of orthographically similar items. Presentation of words and pictures in the orthographic condition was identical to the homophone condition (see earlier for description). However, here for the target word (e.g., boat'), an orthographically similar distractor was also presented (e.g., as 'boot' in this case (see Fig 5). In addition, a semantic distractor (e.g., anchor) and an unrelated distractor (e.g., mug) were presented. Each orthographically similar word pair were of the same length, differed only by one letter (not first letter) and had similar word frequency (e.g., boat/boot). A paired sample t-test showed that there was not a significant difference in word frequency between orthographically similar pairs $(t = 1.539, df (19), p = .140)$. Nonetheless, we created two lists so participants either saw 'boat' or 'boot' as the target item depending on which list was presented to them. As in the homophonic condition, semantic distractors were adapted when the target item was swapped.

**Procedure.** See procedure outlined in Experiment 1a for description of experiment set up (Fig 2). There were four practice trials and 36 experimental trials. Participants were instructed to click on the picture that they felt was the closest match to the item presented to them.

**Data analyses.** The same analyses as in the first two experiments (1a and 1b) were performed for 100 ms bins starting from 0ms and ending at 2000 ms, including Bonferroni correction for multiple comparisons in the time-bin analyses. In the analyses for the homophone condition, target items, semantic and phonological (homophones of the target word) distractors were compared to unrelated items. For the orthographic condition, target items, semantic and orthographically similar distractors were compared to unrelated items. Trials with incorrect responses (in which the wrong target items were selected) were not included in data analysis.

### Results.

### Homophone condition

Word trials with incorrect responses (in which the wrong target items were selected) were not included in data analysis, making up 8% of the total data.

*Fixations to target items.* As can be seen in Fig 6, both groups had more fixations on the target picture than any other distractors. There was a significant effect of target on first to fourth polynomials *(all ps<0.05)*. There was no significant effect of group *(Estimate=.00, SE=.05, p=0.96)*. There were no interactions between group or any of the time polynomials (See Table 10).

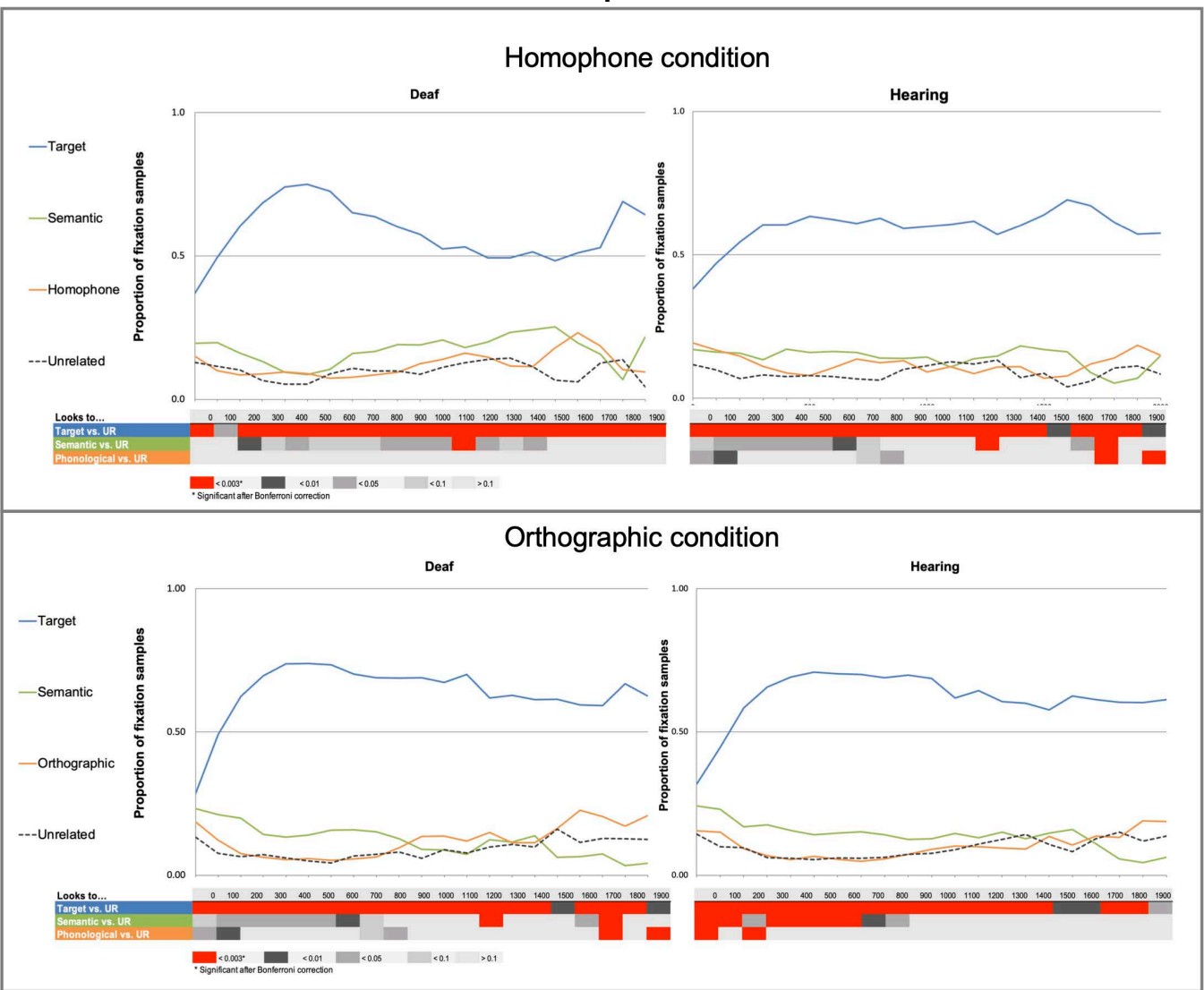

**Fig 6. Results from the homophone (a) and orthographic (b) conditions.** Graphs display the proportion of fixation from the first bin 0−100 ms to the last bin 1900−200 ms for deaf and hearing readers in each condition. Summary tables below the graphs show significant differences in the bin-by-bin analyses between the unrelated condition and each of the remaining distractor conditions: target (T), semantically related (S), and phonologically or orthographically related for hearing and deaf participants separately for word and pseudowords.

The bin-by-bin GLMM analyses showed that deaf and hearing participants performed similarly when comparing target and unrelated items in the homophone condition, with fixations on target items being significant after Bonferroni corrections in almost all time-windows from 0 to 2000 ms: deaf 0–1500 and 1700 ms to 2000, hearing 0–1500 and 1600–1900 ms.

*Fixations to semantic distractors.* Overall, there was a significant effect of semantics on the first polynomial *(Estimate = −0.91 SE = 0.31, p < 0.00)*. There was no significant effect of group *(Estimate = −0.01, SE = 0.02, p = 0.59)*. There were no interactions between group or any of the time polynomials (See Table 10). The bin-by-bin GLMM analyses showed that fixations on semantically related items were significant for the hearing group at 1200–1300 ms and 1700–1800 ms (both *p < 0.001*) and for the deaf group at 1500–1600 ms *p < 0.001)*.

*Fixations to homophonic distractors.* The growth curve analysis did not show any significant effect (See Table 10). The bin-by-bin GLMM analyses showed that for the hearing participants fixations on items that were homophones of the target item were significant after Bonferroni corrections during the 1700–1800 ms and 1900–2000 ms time-window *(*both *p < 0.001)*. Deaf participants did not have fixations on homophones significantly more than unrelated items during any time-window throughout all trials.

## Orthographic condition

Word trials with incorrect responses (in which the wrong target items were selected) were not included in data analysis, making up 3% of the total data. As can be seen in Fig 6, both groups had fixations to the target picture more often than any other distractors and analyses showed no significant differences between deaf and hearing participants. Both groups were also more drawn to semantic distractors in comparison to unrelated items up to about 700 ms.

*Fixations to target items.* Fig 6 shows that deaf and hearing participants' fixations on target items were higher than for unrelated items. There was a significant effect of target on the first to fourth polynomials *(all ps < 0.01)*. There was no significant effect of group *(Estimate = .05, SE = .05, p = 0.35)*. There were no interactions between group and any of the time polynomials (See Table 11). The bin-by-bin GLMM analyses showed that deaf and hearing participants performed similarly when comparing target and unrelated items in the orthographic condition, with fixations on target items being significant after Bonferroni corrections in almost all time-windows from 0 to 2000 ms: deaf 0–1600 ms and 1800–1900 ms, hearing 0–1500 ms and 1700–1900 ms.

*Fixations to semantic distractors.* There was a significant effect of semantics on the first polynomial *(Estimate = −0.94, SE = 0.30, p < 0.00)*. There was no significant effect of group *(Estimate = −0.01, SE = 0.02, p = 0.59)*. There were no

**Table 10. Parameter estimates for each time polynomial, group and interactions for each time polynomial and group for target items, semantic and homophonic distractors in the homophone condition.**

| Target Items | | | | | Semantic Distractors | | | | Homophonic Distractors | | | |
|---|---|---|---|---|---|---|---|---|---|---|---|---|
| **Fixed Effects** | Estimate | SE | T value | P value | **Estimate** | SE | T value | **P value** | **Estimate** | SE | T value | P value |
| **Intercept** | 0.56 | 0.03 | 21.14 | <0.001 | 0.05 | 0.01 | 5.01 | <0.00 | 1.61 | 6.39 | 2.52 | 0.02 |
| **Time polynomial 1** | 1.29 | 0.56 | 2.32 | **0.03** | −0.91 | 0.31 | −2.98 | **<0.001** | −5.29 | 2.34 | −0.23 | 0.82 |
| **Time polynomial 2** | −1.30 | 0.45 | −2.83 | **0.01** | −0.01 | 0.17 | −0.04 | 0.97 | 1.86 | 2.30 | 0.81 | 0.42 |
| **Time polynomial 3** | 1.93 | 0.55 | 3.54 | **<0.001** | 0.15 | 0.17 | 0.91 | 0.36 | −4.71 | 1.64 | −0.29 | 0.77 |
| **Time polynomial 4** | −0.91 | 0.37 | −2.46 | **0.02** | 0.04 | 0.17 | 0.23 | 0.82 | 1.08 | 1.62 | 0.67 | 0.51 |
| **Group** | 0.00 | 0.05 | 0.05 | 0.96 | −0.01 | 0.02 | −0.54 | 0.59 | −6.46 | 1.28 | −0.01 | 1.00 |
| **Time polynomial 1 x Group** | 1.13 | 1.11 | 1.01 | 0.32 | −0.11 | 0.61 | −0.19 | 0.85 | −1.11 | 4.68 | −0.237 | 0.81 |
| **Time polynomial 2 x Group** | −0.19 | 0.92 | −0.21 | 0.83 | 0.36 | 0.33 | 1.06 | 0.29 | 6.15 | 4.59 | 1.34 | 0.19 |
| **Time polynomial 3 x Group** | −1.71 | 1.09 | −1.57 | 0.12 | 0.33 | 0.33 | 1.00 | 0.32 | 5.38 | 3.27 | 1.65 | 0.10 |
| **Time polynomial 4 x Group** | −0.98 | 0.74 | −1.33 | 0.19 | −0.54 | 0.33 | −1.621 | 0.10 | −6.38 | 3.25 | −0.20 | 0.84 |

**Table 11. Parameter estimates for each time polynomial, group and interactions for each time polynomial and group for target items, semantic and orthographic distractors in the orthographic condition.**

| Target Items | | | | | Semantic Distractors | | | | Orthographic Distractors | | | |
|---|---|---|---|---|---|---|---|---|---|---|---|---|
| **Fixed Effects** | Estimate | SE | T value | P value | **Estimate** | SE | T value | P value | **Estimate** | SE | T value | P value |
| **Intercept** | 0.58 | 0.03 | 22.96 | <0.00 | 0.05 | 0.01 | 5.19 | <0.00 | 0.02 | 0.01 | 2.68 | 0.01 |
| **Time polynomial 1** | 1.54 | 0.54 | 2.83 | **0.01** | −0.94 | 0.30 | −3.14 | **0.00** | −0.02 | 0.23 | −0.07 | 0.95 |
| **Time polynomial 2** | −1.36 | 0.46 | −2.98 | **0.01** | −0.00 | 0.17 | −0.2 | 0.98 | 0.23 | 0.23 | 1.00 | 0.32 |
| **Time polynomial 3** | 1.95 | 0.53 | 3.67 | **0.00** | 0.15 | 0.17 | 0.92 | 0.36 | 0.01 | 0.28 | 0.03 | 0.97 |
| **Time polynomial 4** | −0.87 | 0.36 | −2.38 | **0.02** | 0.04 | 0.17 | 0.24 | 0.80 | 0.15 | 0.27 | 0.57 | 0.57 |
| **Group** | 0.05 | 0.05 | 0.93 | 0.36 | −0.01 | 0.02 | −0.55 | 0.59 | −0.00 | 0.01 | −0.11 | 0.91 |
| **Time polynomial 1 x Group** | 1.62 | 1.09 | 1.49 | 0.15 | −0.16 | 0.60 | −0.27 | 0.79 | −0.14 | 0.47 | −0.31 | 0.76 |
| **Time polynomial 2 x Group** | −0.31 | 0.91 | −0.34 | 0.74 | 0.36 | 0.34 | 1.08 | 0.28 | 0.59 | 0.46 | 1.27 | 0.21 |
| **Time polynomial 3 x Group** | −1.68 | 1.06 | −1.58 | 0.12 | 0.34 | 0.33 | 1.02 | 0.31 | 0.54 | 0.55 | 0.98 | 0.34 |
| **Time polynomial 4 x Group** | −0.90 | 0.73 | −1.24 | 0.22 | −0.53 | 0.33 | −1.60 | 0.11 | −0.06 | 0.54 | −0.12 | 0.91 |

interactions between group and any of the time polynomials (See Table 11). The bin-by-bin GLMM analyses showed that fixations on semantically related distractors were significant during the 1200–1300 and 1700–1900 ms time-window for deaf participants while they were much earlier and extended in time for the hearing participants: significant from 0 to 200 ms, and 300–700 ms.

*Fixations to orthographically similar distractors* In the growth curve analyses there were no effects of orthography, group nor interactions between group and any of the time polynomials (See Table 11). The bin-by-bin GLMM analyses showed that for deaf participants the fixations on orthographically related distractors were significant after multiple comparisons correction only from 1700 to 1800ms and 1900–2000 ms. Besides in this later period, the proportion of fixations on orthographic distractors was lower than on unrelated distractors. For hearing participants, the looks to orthographic distractors were significantly higher than to unrelated distractors from 0 to 100 ms, and 200–300 ms.

## Discussion – Experiment 2

In the homophone and orthographic conditions, there was very little difference in fixations on target items between deaf and hearing participants. This replicates results from Experiment 1a.

Deaf and hearing readers had a larger percentage of fixations on semantically related than unrelated distractors. While processing seems similar between deaf and hearing readers, there were small but noteable differences in the specific time course of this effect. In the homophone condition deaf readers had more fixations on semantic related than unrelated distractors but the differences seemed restricted to one specific late time window. The hearing readers had slightly longer effects. In the orthographic condition deaf readers only had fixations on semantic related distractors during the 1200 and the 1700–1800 ms time-windows. Hearing readers had fixations on semantic related distractors in most time-windows between 0 and 700 ms.

In line with the results from Experiment 1a, the profile of looks to the phonological distractor differed between the two groups: hearing participants had fixations more often on the phonological distractors between 1700–2000 ms. This late pattern of fixations may be interpreted in terms of the participant rechecking the pictures as hearing participants also continued to have fixations on the target picture during these time-windows. For the deaf readers, we did not see more fixations on homophones than unrelated distractors in any of the time-windows. In this experiment participants were required to make a choice for all trials, but the target picture was always present hence making phonological information less relevant than in Experiment 1b. That is, participants could recognise the word and click on the target picture either using the phonological information or via the direct orthography to semantics route. Therefore, the looks to the phonological and

unrelated distractors are low. Even with low proportions of fixations, the effect is significant for hearing readers in these later time windows, likely representing a robust influence of phonology during processing. These subtle differences provide some evidence that deaf readers are less likely to be influenced by or make use of phonological information during word recognition if the phonological information is not required by the task.

Again, even with a low proportion of fixations to the orthographic distractors, both groups showed an effect of orthography during processing at different restricted time windows. Hearing readers had fixations on orthographic distractors between 0 and 300 ms, whereas deaf readers' fixations on orthographic distractors occurred much later at 1900–2000 ms. Besides these later effects were in the opposite direction to the early effects observed in hearing readers. That is, deaf readers looked less at the orthographic than to the unrelated distractors. This may be because of less competition and greater precision for deaf readers compared to hearing readers when activating orthographic representation of the word, i.e., deaf readers are less distracted by orthographically similar words, and are able to quickly identify the target word. This finding is in line with results reported by Meade et al. [48] who found that phonological processing is not necessary for orthographic precision. Deaf readers have been found to be faster and more accurate in lexical decisions tasks which provides support for less competition from distracters and greater orthographic precision compared to hearing readers [13,29]. Further research is needed to establish whether and how differences in tasks affect the time-course of orthographic processing.

## General discussion

The two overarching goals of the work reported here were: (1) to investigate the time-course of orthographic, semantic, and phonological activation during single word recognition during tasks that favour semantic processing; (2) To investigate the role of phonology during single word recognition as a function of task demands. Deaf individuals have different language experiences from hearing readers. In particular, the deaf readers involved in the current studies are mostly early/native signers of BSL, for whom English is their second/additional language. Their language experience may influence how orthographic information links to semantic and phonological information, as well as the time-course of activation. Bringing together the two main goals of this work, our research provides new insights into word recognition in deaf readers.

Our results show that deaf skilled readers and matched hearing readers show a similar pattern of fixations on target and semantic distractor pictures when processing target words across all experiments. Experiment 1b further showed that both groups were able to select a target picture (corresponding to a pseudohomophone) following a similar pattern of fixations. Again, both groups showed largely similar semantic activation, with some differences in timings, across our three experiments. While group comparisons did not show differences in the way the two groups processed phonological information, separate bin-by-bin analyses showed that within each group there were differences in processing of phonological information in several conditions across all experiments.

Taken together, the experiments reported here show that deaf readers extract semantic information from words following a similar time-course to hearing readers despite differences in language experiences. In addition, our studies demonstrate that the visual world paradigm can be used successfully to explore the time-course of activation of orthographic, semantic, and phonological information in visual word recognition of deaf and hearing readers, providing novel insight in the time-course activation of semantic and phonological information in the two groups. Furthermore, the findings here highlight the importance of studying the time course of processing as subtle time course differences, such as the observed here, are unlikely to be captured by a single measure such as accuracy, at the final stage of processing. However, in future, such studies should not include target items to encourage further looks to semantic and phonological distractors without the distraction of target items and should control highly for visual properties of the picture stimuli. Furthermore, it would be worth replicating this study with a larger group of deaf participants to increase power. These may lead to more robust results.

## What is the role of phonology for deaf skilled readers?

There were subtle differences in the way deaf and hearing readers processed phonological information during the visual word processing tasks. In Experiment 1a, hearing readers had higher fixations on the picture that corresponded to the word from which the pseudohomophone was derived, compared to unrelated items, (e.g., coat/KOTE as the pseudotarget). However, this was not the case for deaf readers. In Experiment 1b, when presented with a pseudohomophone nonword, hearing readers had fixations on pseudosemantic more than unrelated distractors in several time-windows. In contrast, deaf readers only had fixations on pseudosemantic distractors more than unrelated items in the 1100–1200 ms time-window. Finally in Experiment 2, only hearing readers had higher fixations on homophonic distractors than on unrelated items in some time-windows.

While these results must be taken with some caution because only a handful of significant group differences were found in different time bins, we see consistency in the tasks and experiments to show that phonological codes are activated less frequently during reading by deaf skilled readers compared to hearing readers. For hearing readers, phonological processing seems to be automatic and essential, whether it is a task requirement or not [18]. This finding is in line with Glezer et al., (2018) who found that compared to hearing readers, deaf skilled readers showed weaker phonology selectivity. However, they showed similar orthography selectivity, which led to Glezer et al. [28] reaching their conclusion that while deaf readers do activate phonology when making phonological decisions, their phonological representations are more coarse and less precise but without any impact on the neural network associated with skilled reading. To summarise, deaf readers can use phonology to complete some tasks, but it is not automatic nor is it a marker of skilled reading in deaf adults.

Crucially, although the visual world experiments show some indication of reduced phonological activation in deaf readers, this factor did not appear to have an impact on reading abilities as reading levels were matched between deaf and hearing participants. Furthermore, even with differences in language experiences the activation of semantic information from orthography did not differ (i.e., deaf readers and hearing readers' activation of semantic information from orthography followed a similar time-course).

## The role of phonology for hearing readers (and how it compares to deaf readers)

Theoretical proposals of reading in hearing adults who are skilled readers assume the interplay between activation of orthographic, semantic, and phonological codes [18,19,49,50]. The general assumption is that phonological activation arises from phonological decoding processes, namely, the operation by which beginning hearing readers recover the pronunciation of any pronounceable string of letters by using knowledge about the associations between phonemes and graphemes [51]. Hearing readers already have phonological representations for spoken words and processing those phonological representations can support the activation of semantic codes. For adult hearing readers, activation of phonological codes can support their visual word recognition in two ways. Firstly, because phonological codes are connected to orthographic codes, activation of phonological codes can enhance orthographic (visual) recognition as suggested by Meade et al. [48]. Secondly, because phonological codes are linked to semantics in hearing readers, it can enhance activation of semantic information about the word, thus enhancing the processing of its meaning.

Reading instruction in the UK typically uses speech-based phonics focussed approaches for both deaf and hearing children. Nevertheless, reduced access to speech-based phonology may mean that deaf readers may develop phonological codes in a different manner. One primary source of information from which deaf people can derive phonological information is from speechreading [52–55]. That is by extracting lexical and sublexical level information (e.g., phonological information) from visual information (also referred to as visemes) associated with words [52]. Studies have shown that speechreading skills develop over time [56]. It has also been shown that increased speechreading skill is likely to enable deaf children to further access speech-based phonology [54,55]. This may explain why we see some evidence of phonological processing amongst participants in this study. Kyle & Harris [10] found that deaf children's phonological awareness improved as they were learning to read, which suggests that deaf children may also gain access to speech-based

phonology via print as well as speechreading. It is also worth noting that sign-based and fingerspelling-based phonological awareness has been shown to be related to printed word reading skills in deaf children [57,58]. The question here is to what extent do phonological codes based on non-auditory information support reading in adult deaf skilled readers?

Our studies, along with a few other [2,23,28], show that there are differences in the way deaf and hearing adults who are skilled readers process phonology. Notably, deaf readers' responses to pseudohomophones differed in experiments 1a and 1b, as the task requirements changed. In experiment 1b, deaf readers accessed the target word using pseudo-homophones. This suggests that deaf readers may have alternative ways to using phonology to decide whether a letter string is a word or not and that phonology may not be as useful for them compared to hearing readers. In an ERP study by Gutierrez-Sigut et al. [2], the N250 activation (related to early and automatic use of sub-lexical phonological components) was comparable in deaf and hearing readers, showing that deaf readers can activate phonological codes during a lexical decision task. Interestingly, the deaf readers had subtle differences in the later N400 component, which is related to activation of whole-word forms and their meanings. Specifically, deaf readers showed reduced negativities in left electrodes than the hearing readers, which may indicate that deaf readers use different neural pathways or strategies for reading. This pattern of findings suggests that even if deaf readers might activate phonological codes automatically, they do not always link them up to semantic codes as strongly as hearing readers do [2] because as explained above, deaf individuals may access phonological codes later compared to hearing readers (i.e., while learning to read). These differences between deaf and hearing readers are in line with the findings reported in this study, i.e., deaf readers utilise phonological information less frequently than hearing readers do.

Given Seidenberg & McClelland's [59] triangle model of reading, the difference between deaf and hearing readers' phonological information processing can be accounted for in terms of lack of (or weaker) activation from phonological forms to semantics. This is likely to be because deaf readers' phonological representations of spoken language are less specified than for hearing individuals. The deaf readers in this study consisted mainly of native and near-native signers who used BSL on a daily basis, which may mean they are not activating semantic information from spoken phonological information (via speechreading) as often as hearing readers who use spoken language on a daily basis. Additionally, many of the deaf readers are likely to have acquired English as a second language, which could be another factor in why the connections between spoken language phonology and semantics are weak. Similar results have been observed in hearing bilinguals, where their performance on phonological tasks in their second language (L2) suggests a less active semantic network in the L2 compared to their first language (L1) [60].

If successful reading can be achieved despite impoverished access to phonology, why do the connections between semantics and phonology exist for hearing readers? For hearing readers, these connections exist as they were present prior to learning to read (mapping between spoken word form and spoken word meaning) and orthographic to semantic mappings were formed via those existing phonological codes. It is possible that the connections seen between phonology and semantics during word recognition in hearing readers may be 'left-over' effects from their learning history (i.e., learning to map between orthography and semantic codes via phonological codes). Alternatively, these phonological codes may increase semantic activation and this information may make the retrieval of semantic information more secure [61].

Prior to learning to read, it is possible that most deaf readers in this study developed connections between sign language and semantics (linking up sign forms to concepts), which they subsequently mapped onto print [13,62,63]. Early access to language has enabled them to develop robust connections between orthographic and semantic codes and become successful readers.

## Conclusion

The present study has led to two important conclusions. First, although there are some differences in word processing by deaf and hearing people, the two groups showed important similarities in the activation of orthographic and semantic codes. Second, unlike hearing readers, deaf readers did not use phonological information *automatically* to access

semantics during single word reading. Despite this, reading skill was not impacted. However, it is important to note that this study explored deaf adults who were skilled readers. What is unknown is how they became skilled readers and to what extent speech-based phonological information was used during reading acquisition. These results open up a number of other questions relating to strategies that deaf skilled readers may employ to overcome the lack of phonological activation. A potentially informative approach to answer these questions could be to develop population-level studies that capitalise on the visual world paradigm to explore word processing in deaf and hearing readers with different levels of reading skills, age, and multilingual experiences. Understanding crucial differences in the reading processes of these groups in their learning trajectories will help inform theories of reading and teaching practices. This, in turn, will improve literacy attainment in the deaf population.

## Supporting information

**S1 Appendix. Stimuli list for Experiment 1a and 1b.**
(DOCX)

**S2 Appendix. Stimuli list for Experiment 2.**
(DOCX)

## Author contributions

**Conceptualization:** Katherine Rowley, Mairead MacSweeney, Gabriella Vigliocco.

**Formal analysis:** Katherine Rowley, Eva Gutierrez-Sigut.

**Funding acquisition:** Katherine Rowley.

**Investigation:** Katherine Rowley.

**Methodology:** Katherine Rowley, Mairead MacSweeney, Gabriella Vigliocco.

**Supervision:** Mairead MacSweeney, Gabriella Vigliocco.

**Writing – original draft:** Katherine Rowley.

**Writing – review & editing:** Eva Gutierrez-Sigut, Mairead MacSweeney, Gabriella Vigliocco.

AcknowledgmentKatherine Rowley, Gabriella Vigliocco and Mairéad MacSweeney conceptualised the study and designed the experiments. Katherine Rowley conducted the experiments, collected and analysed the data. Eva Gutierrez-Sigut analysed the data. Katherine Rowley wrote the initial manuscript draft. Mairéad MacSweeney, Eva Gutierrez-Sigut and Gabriella Vigliocco provided critical revisions. All authors reviewed and approved the final manuscript.

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
