## [Decision Letter · Decision Letter 0]

7 Jan 2025

Dear Dr. Rowley,

We look forward to receiving your revised manuscript.

Kind regards,

Mary Diane Clark, PhD

Academic Editor

PLOS ONE

Additional Editor Comments:

Nicely done paper---I still have a hunch that phonology is serving as an independent code not related to deaf readers--a study for another day. The reviewers have some important points for your to consider in your revision.

I have a few things including that citations are not done in APA for please check the website for the correct format.

Some minor issues:

line 247 showed an array. (delete the word with)

1a you use first, second etc. polynomials and 1b you. Use 1, 2, 3

Please be consistent

Can include some citations in your discussions for each experiment especially in 1a and 1b?

Reviewers' comments:

Reviewer's Responses to Questions

**Comments to the Author**

1. Is the manuscript technically sound, and do the data support the conclusions?

Reviewer #1: Yes

Reviewer #2: Yes

2. Has the statistical analysis been performed appropriately and rigorously?

Reviewer #1: Yes

Reviewer #2: Yes

3. Have the authors made all data underlying the findings in their manuscript fully available?

Reviewer #1: Yes

Reviewer #2: No

4. Is the manuscript presented in an intelligible fashion and written in standard English?

Reviewer #1: Yes

Reviewer #2: Yes

Reviewer #1: The authors present a set of eye tracking studies using the Visual World Paradigm to investigate orthographic, phonological, and semantic processing in skilled deaf and hearing readers. Although some of the findings are not particularly robust, one major strength of these studies is their focus on task demands, which helps cut through the noise and make sense of the mixed results in the literature about phonological processing for deaf readers. The manuscript could be enhanced with greater methodological detail and some expansion of the discussion, but overall, I believe it makes a helpful contribution to the field and would be of interest to the readership.

Major comments

• The manuscript is clear, organized, and well written. The introduction thoroughly references relevant literature and provides clear motivation for the studies. The systematic reporting of results was easy to follow, and details for each study were balanced with integration of findings across studies. Addressing some run-on sentences and minor copy edits will further improve readability.

• I would like to see more details about the participants. I am glad to see that groups are carefully matched with respect to age, reading ability, and gender. Did the authors collect any other measures of relevant skills (e.g., English phonological awareness, spelling, vocabulary)? I would be curious to how such behavioral measures relate to the eye tracking effects.

• I would also like more information about the participants’ hearing history and amplification. I assume that the degree of deafness reported is unaided. Do you have information about aided thresholds, given that the majority of participants use hearing aids or cochlear implants? Other relevant details may include: age of implantation, years of use, daily use. Without a measure of phonological awareness, it will be even more important to quantify how much access the participants have to auditory input. Similarly, I see that the late signers all had 10+ years of BSL, but age of BSL exposure would indicate whether language deprivation may have influenced the results.

• Was a power analysis conducted? The authors acknowledge that future studies should not include target items to promote looks to distractors, but I am concerned that the sample size, number of items, high probability of fixating the target compared to a competitor, and exclusion of trials with incorrect responses (especially high for Experiment 1b) have impacted power.

• The discussion (paragraph starting on line 1028) could be expanded by discussing differences in instruction for deaf and hearing readers that affect how deaf readers develop speech-based phonological representations and integrate them with orthographic and semantic information. I think it is also worth mentioning that deaf readers can also develop sign-based phonological awareness, which also supports reading development and complements speech-based phonological awareness for those who can access both sign and speech. See Lederberg, A. R., Branum-Martin, L., Webb, M. Y., Schick, B., Antia, S., Easterbrooks, S. R., & Connor, C. M. (2019). Modality and interrelations among language, reading, spoken phonological awareness, and fingerspelling. The Journal of Deaf Studies and Deaf Education, 24(4), 408-423.

• The figures are too small. It is difficult to read the text and to make out some of the picture stimuli.

Minor comments

• Line 96. This section does a nice job of describing studies that had explicit vs. implicit phonological processing demands, which sets up the finding that deaf readers can and do activate phonology for tasks that require it but do not automatically do so for tasks that do not. Does the type of phonological task matter? The studies reviewed here use a variety of deep (phonemic awareness) and shallow (alliteration, rhyming, syllable counting) phonological awareness tasks.

• Line 361. How were pictures selected? Were they balanced on visual properties like color, contrast, brightness, etc. that may influence their visual saliency and attract the participants’ gaze?

• Line 457. Table 5. Is there a reason why the 0.13 p value is highlighted?

• Line 474 typo: fourth

• Line 597. Table 8. Please repeat the note explaining the * for near-native signers

• Line 624. This error rate seems high, especially compared to the other two experiments. Any explanation or comment on how this may have affected power?

• Line 680. Delete ‘on’

• Line 718-719. I would specify that the “processing of words is very similar between skilled deaf and hearing readers” when phonological processing is required for the task. This finding contrasts with the others showing that deaf and hearing readers do not always use phonology similarly.

• Line 768. The examples of stimuli provided here (e.g., night, knight, day, chocolate) made it easy to keep track of the different conditions. Could the same be provided for the other experiments?

• Line 892. Typo: replicates

• Line 1008. Close parentheses

• Line 1068. This sentence is confusing. The punctuation makes it difficult to parse and comprehend.

Reviewer #2: Summary

The study investigates orthographic, semantic, and phonological processing during single word reading in 20 deaf and 20 hearing adult readers, matched on reading skill. Eye-movements were tracked using an adaptation of the visual world paradigm, focusing on implicit phonological processing. The study shows that both deaf and hearing readers exhibit similar time-courses for activating orthographic and semantic information. However, deaf readers rely more on orthographic and semantic cues rather than phonological information. When the task required explicit phonological processing (Experiment 1b), deaf readers could use phonological information effectively. These findings suggest that while phonological processing is less utilized by deaf readers, it does not hinder their reading abilities. This is the first study to use the visual world paradigm to examine word processing in deaf readers. I commend the authors on tackling this important topic.

Strengths

First, the use of the visual world paradigm to examine word processing in deaf readers is novel and provides valuable insights into the reading processes of this population. Secondly, the careful matching of deaf and hearing participants on reading skill ensures that differences observed are not due to disparities in reading ability. Finally, the study provides a detailed analysis of orthographic, semantic, and phonological processing, contributing to our understanding of how deaf readers process written language.

Concerns and suggestions for improvement

First of all, the authors should provide the age of exposure to sign language for late signers. This information is crucial for understanding the variability in language acquisition and its impact on reading processes.

Second, based on the information provided in the figures, I noticed that some (target) images could be more complex than foils. For example, the picture of a boat seems quite complex while the picture of the orthographic distractor is less complex. This could lead to discrepant fixation times across conditions. Without seeing the rest of the picture stimuli, I can’t verify this, of course. The authors should consider reporting a measure of visual complexity to ensure that differences in fixation times are not due to image complexity.

Thirdly, I recommend the authors clarify the time range for the first, second, third, etc., polynomials. Is it 500-1000ms, 1000-1500ms, etc.? Providing this information would help in understanding the temporal dynamics of the observed effects.

The authors should clarify the motivation for the experiments, particularly Experiment 1b, which seemed a bit ad hoc and lack a clear justification. The authors should add a discussion of what is meant by “different viewing strategies” and how these strategies are expected to influence the results.

I would also like to encourage the authors to elaborate more on why it was important to include a mixture of homophonic and non-homophonic nonwords. This explanation would benefit the audience who are less familiar with how deaf people might process written words.

I also felt that clear hypotheses or predictions for Experiments 1a and 1b were missing. Providing explicit hypotheses would strengthen the theoretical framework of the study and guide the interpretation of the results.

Were the deaf participants aware of pseudohomophones in Study 1B? In Experiment 1b, it is important to know whether the participants figured out that the words were pseudohomophones. If they did, they could have used a more overt phonological coding than those who treated the words as non-homophonic pseudowords or nonwords. This could significantly impact the results and their interpretation.

Interpretation of Findings: The differences between groups across conditions are too subtle, yet the authors argue there are key differences in how deaf and hearing readers manage visual information. This argument seems unsubstantiated. A more nuanced interpretation of the findings is needed, acknowledging the subtlety of the observed differences.

My final comment relates to critical discussion of the visual world paradigm. As this appears to be the first study that has used the visual world paradigm to analyze word processing in deaf readers, a more critical discussion of this method is warranted. Previous work suggests that deaf readers exhibit differences in how they manage visual information, especially in the parafovea. The authors should discuss how the visual world paradigm addresses or fails to address these differences.

Overall, this study makes a significant contribution to our understanding of reading processes in deaf readers. However, addressing the above weaknesses would strengthen the study’s methodology and interpretation of findings. I commend the authors for their innovative approach and encourage them to consider these suggestions for improvement.

**Do you want your identity to be public for this peer review?** For information about this choice, including consent withdrawal, please see our Privacy Policy

Reviewer #1: No

Reviewer #2: **Yes: ** Zed Sehyr

---

## [Author Response · Author response to Decision Letter 1]

14 May 2025

Editor

Nicely done paper---I still have a hunch that phonology is serving as an independent code not related to deaf readers--a study for another day. The reviewers have some important points for you to consider in your revision.

Thank you, we are glad you liked the paper. We have done our best to address all of the reviewer comments.

I have a few things including that citations are not done in APA for please check the website for the correct format.

I have checked through all the citations and references making sure they are in APA format.

line 247 showed an error (delete the word with). Well spotted, thank you, ‘with’ has been deleted. (pg 11, line 267)

1a you use first, second etc. polynomials and 1b you. use 1, 2, 3. Please be consistent Thank you, I have replaced 1, 2, 3 etc with first, second etc.

Can include some citations in your discussions for each experiment especially in 1a and 1b? I have added citations in 1a. (pg 29, line 631)

This finding is in line with other studies that did not find an effect of phonology in deaf skilled readers during single word recognition (e.g., Belanger et al., 2012; Farina et al., 2017).

And in 1b. (pg 38, lines 831-833)

This finding is in line with some other studies that show deaf readers can and do process phonology but less so compared to hearing readers (e.g., Glezer et al., 2018).

Reviewer 1

I would like to see more details about the participants. I am glad to see that groups are carefully matched with respect to age, reading ability, and gender. Did the authors collect any other measures of relevant skills (e.g., English phonological awareness, spelling, vocabulary)? I would be curious to how such behavioral measures relate to the eye tracking effects.

We agree, it would have been interesting to look at other relevant skills. However, since the additional measures suggested were not part of our research questions for this study, we did not collect these data.

I would also like more information about the participants’ hearing history and amplification. I assume that the degree of deafness reported is unaided. Do you have information about aided thresholds, given that the majority of participants use hearing aids or cochlear implants? Other relevant details may include age of implantation, years of use, daily use. Without a measure of phonological awareness, it will be even more important to quantify how much access the participants have to auditory input. Similarly, I see that the late signers all had 10+ years of BSL, but age of BSL exposure would indicate whether language deprivation may have influenced the results. Yes, the level of deafness is unaided. We have added ‘unaided’ to the description to make this explicit.

We agree this additional information would have been informative in describing the participants, however we did not collect this information.

For most severely to profoundly deaf people hearing technology does not provide them with clear access to speech. This is particularly true for deaf signers who chose to use a sign language as their primary language.

We were careful to recruit deaf signers that do not have clear access to speech (i.e., they are unable to understand speech without additional visual information). We have added this information to the participant’s description: “All participants reported that they were unable to have clear access to speech through a hearing aid” (pg 16, lines 362-364)

There were very few late learners of BSL (3/20), they were skilled readers, and they were matched to the hearing participants for reading level and thus unlikely to have experienced language deprivation. We have added to the text that their first age of Exposure to BSL was 11yrs (secondary school).

Was a power analysis conducted? The authors acknowledge that future studies should not include target items to promote looks to distractors, but I am concerned that the sample size, number of items, high probability of fixating the target compared to a competitor, and exclusion of trials with incorrect responses (especially high for Experiment 1b) have impacted power. Unfortunately, a power analysis was not conducted for this study, for which the data were collected over 7 yrs ago. Nevertheless, our sample size aligns with previous studies of reading conducted with this difficult to reach population.

We have added a line to the participants section, ‘A power analysis was not conducted, as this study concerns a difficult to reach population, however our sample size aligns with previous studies (Farina et al., 2017; Gutierrez-Sigut et al., 2018)’. (pg 14, 348-350).

And to the general discussion, ‘Furthermore, it would be worth replicating this study with a larger group of deaf participants to increase power.’ (pg 51, line 1111-1113)

We have also added a discussion of the high error rate in experiment 1b to the discussion section (pg 36, line 773-778).

‘The error rate in this study was high (21%). It is likely that this is due to the nature of the task. Participants were required to select the picture that best matched the presented pseudohomophone, without encountering any real words during the experiment. Since pseudohomophones are not actual words and the task demanded quick responses, this likely contributed to the high error rate.’

The discussion (paragraph starting on line 1028) could be expanded by discussing differences in instruction for deaf and hearing readers that affect how deaf readers develop speech-based phonological representations and integrate them with orthographic and semantic information. I think it is also worth mentioning that deaf readers can also develop sign-based phonological awareness, which also supports reading development and complements speech-based phonological awareness for those who can access both sign and speech. See Lederberg, A. R., Branum-Martin, L., Webb, M. Y., Schick, B., Antia, S., Easterbrooks, S. R., & Connor, C. M. (2019). Modality and interrelations among language, reading, spoken phonological awareness, and fingerspelling. The Journal of Deaf Studies and Deaf Education, 24(4), 408-423. In relation to differing instruction – in the U.K., instruction does not really differ much between deaf and hearing children.

We have added a bit more information in this section (pg 53, line 1173-1176).

‘Reading instruction in the UK typically uses speech-based phonics focussed approaches for both deaf and hearing children. Nevertheless, reduced access to speech-based phonology may mean that deaf readers may develop phonological codes in a different manner.’

And the below on pg 53, line 1181-1188.

‘Studies have shown that speechreading skills develop over time (Kyle et al., 2013). It has also been shown that increased speechreading skill is likely to enable deaf children to further access speech-based phonology (Pimperton et al., 2019; Buchanan-Worster, et al., 2020). This may explain why we see some evidence of phonological processing amongst participants in this study. Kyle & Harris (2010) found that deaf children’s phonological awareness improved as they were learning to read, which suggests that deaf children may also gain access to speech-based phonology via print as well as speechreading.’

With regard to sign phonological awareness, thank you for raising this. We have added (pg 53, line 1188-1191):

It is also worth noting that sign-based and fingerspelling-based phonological awareness has been shown to be related to printed word reading skills in deaf children (Holmer et al., 2016; Lederberg et al. (2019).

The figures are too small. It is difficult to read the text and to make out some of the picture stimuli. Fixed.

Line 96. This section does a nice job of describing studies that had explicit vs. implicit phonological processing demands, which sets up the finding that deaf readers can and do activate phonology for tasks that require it but do not automatically do so for tasks that do not. Does the type of phonological task matter? The studies reviewed here use a variety of deep (phonemic awareness) and shallow (alliteration, rhyming, syllable counting) phonological awareness tasks. It is a good question. We have added this line to this part (pg 6, line 132), ‘In explicit tasks, different types of phonological awareness are recruited such as shallow (rhyme judgement) or deep (phonemic judgement). However, it seems that there is evidence of phonological awareness amongst deaf readers for both types, e.g., MacSweeney et al’s (2009) study (phonemic awareness) and in a later study (rhyme judgement) (MacSweeney et al., 2013).’

Line 361. How were pictures selected? Were they balanced on visual properties like color, contrast, brightness, etc. that may influence their visual saliency and attract the participants’ gaze? We have added a line to the stimuli description, ‘pictures were taken from clipart.com and were balanced on size and style’. (pg 17, line 383-384)

And the below on pg 17, lines 390-396.

To ensure that visual saliency was evenly distributed across target and distractor images, three participants were asked to identify the image that most captured their attention or to indicate 'none' if no image stood out. On average, image selection ranged from 19.8% to 27.6% across target, semantic, phonological, and unrelated images, while no image was selected in 9.4% of cases. These results suggest that visual saliency was relatively balanced across target and distractor images.

Line 457. Table 5. Is there a reason why the 0.13 p value is highlighted? Thank you for catching this. This has now been removed.

Line 474 typo: fourth Now corrected.

Line 597. Table 8. Please repeat the note explaining the * for near-native signers Done.

Line 624. This error rate seems high, especially compared to the other two experiments. Any explanation or comment on how this may have affected power? We have addressed this point in the response above and also now addressed it in the text pg 36, line 773-778).

Line 680. Delete ‘on’ Deleted.

Line 718-719. I would specify that the “processing of words is very similar between skilled deaf and hearing readers” when phonological processing is required for the task. This finding contrasts with the others showing that deaf and hearing readers do not always use phonology similarly. To improve clarity, we have changed the wording, as we do believe that the combination of Experiment 1 and 2 findings shed light into why previous research offers mixed results. We hope that our argument is now clearer. (pg 38, line 834-845).

‘These results are important, as they show that despite differences in language experience, processing of words is very similar between skilled deaf and hearing readers when the phonological information from words is required to successfully complete a task. This finding is consistent with previous studies that have shown a phonological effect using an explicit task (Emmorey et al., 2013; Glezer et al., 2018) indicating that deaf readers can and do process phonological information when required. Our findings also show that when phonology is not useful to perform the task, as in Experiment 1a, deaf readers seem use it less than hearing readers. The latter finding can explain why some studies using implicit tasks and measuring automatic processing have not found phonological effects in deaf skilled readers (Belanger et al., 2013; Farina et al., 2017)’.

Line 768. The examples of stimuli provided here (e.g., night, knight, day, chocolate) made it easy to keep track of the different conditions. Could the same be provided for the other experiments? This is a great idea. Thank you. We have now added examples of stimuli used to the descriptions of all experiments.

Line 892. Typo: replicates Done.

Line 1008. Close parentheses Added.

Line 1068. This sentence is confusing. The punctuation makes it difficult to parse and comprehend. This sentence has now been revised and hopefully it is now clearer.

Reviewer 2

First of all, the authors should provide the age of exposure to sign language for late signers. This information is crucial for understanding the variability in language acquisition and its impact on reading processes. This information has now been added below the participant tables (pg 16, line 368).

**Late learners used BSL as their main language for 10+ years at the point of testing and age of first exposure was at 11 years of age (secondary school).

Second, based on the information provided in the figures, I noticed that some (target) images could be more complex than foils. For example, the picture of a boat seems quite complex while the picture of the orthographic distractor is less complex. This could lead to discrepant fixation times across conditions. Without seeing the rest of the picture stimuli, I can’t verify this, of course. The authors should consider reporting a measure of visual complexity to ensure that differences in fixation times are not due to image complexity.

Reviewer 1 posed a related question concerning the selection of the images. While image complexity was not matched before the study, the pictures were balanced in terms of size, style and a measure of visual salience (how likely they are to grab attention as indicated by three additional participants). This information is now reported in the paper. (pg 17, lines 383-396).

Thirdly, I recommend the authors clarify the time range for the first, second, third, etc., polynomials. Is it 500-1000ms, 1000-1500ms, etc.? Providing this information would help in understanding the temporal dynamics of the observed effects. We have clarified this and added the information below to the manuscript (pg 21, line 498-503).

‘In the Growth Curve Analyses, the polynomials map to the shape of the function (e.g. linear, quadratic, etc.) (Mirman et al., 2008) rather than to specific time points. Significant results in the GCA would indicate that there are differences over time but does not specify when. Therefore, we followed up with the time bins analysis in order to understand the temporal dynamics better than just observing the differences in the plots.’

The authors should clarify the motivation for the experiments, particularly Experiment 1b, which seemed a bit ad hoc and lack a clear justification. The authors should add a discussion of what is meant by “different viewing strategies” and how these strategies are expected to influence the results. We have further clarified how the experiments map to our research aims (pg 12, lines 295-298).

The two overarching goals of the work reported here are: (1) to investigate the time-course of orthographic, semantic and phonological activation during single word recognition during tasks that favour semantic processing (Experiments 1a and 2); (2) To investigate the role of phonology during single word recognition as a function of task demands (Experiments 1a and 1b).

Below we present the results from three experiments which use a visual world paradigm with different orthographic, semantic and phonological manipulations.

Experiment 1b: Exploring semantic and phonological processing in deaf readers using the visual world paradigm in a forced choice task. (pg 29, line 647).

Regarding viewing strategies, we agree that the reference to ‘viewing strategies’ was unclear and have thus removed it from the paper. (pg 19, line 450-452).

And added, ‘trial order was randomised to reduce the potential impact of participants using different anticipatory strategies for the word or nonword tasks’.

I would also like to encourage the authors to elaborate more on why it was important to include a mixture of homophonic and non-homophonic nonwords. This explanation would benefit the audience who are less familiar with how deaf people might process written words. Thank you for this suggestion. We have added more information on pg 13, line 318.

‘It was important that target nonwords consisted of a mixture of pseudohomophonic (e.g., phly) and non-homophonic (e.g., adge) nonwords for comparison, as this helps to isolate phonological and orthographic processes in word recognition. Pseudohomophones measure the influence of phonology on lexical access, while non-homophonic nonwords

---

## [Decision Letter · Decision Letter 1]

10 Jun 2025

Reading with deaf eyes: automatic activation of speech-based phonology during word recognition is task dependent

PONE-D-24-50427R1

Dear Dr. Rowley,

We’re pleased to inform you that your manuscript has been judged scientifically suitable for publication and will be formally accepted for publication once it meets all outstanding technical requirements.

Kind regards,

Mary Diane Clark, PhD

Academic Editor

PLOS ONE

Additional Editor Comments (optional):

Thank you for responding to the reviewers comments. the paper is a nice extension of past work

Reviewers' comments:

Reviewer's Responses to Questions

**Comments to the Author**

Reviewer #1: All comments have been addressed

Reviewer #3: All comments have been addressed

2. Is the manuscript technically sound, and do the data support the conclusions?

Reviewer #1: Yes

Reviewer #3: Yes

3. Has the statistical analysis been performed appropriately and rigorously?

Reviewer #1: Yes

Reviewer #3: Yes

4. Have the authors made all data underlying the findings in their manuscript fully available?

Reviewer #1: Yes

Reviewer #3: Yes

5. Is the manuscript presented in an intelligible fashion and written in standard English?

Reviewer #1: Yes

Reviewer #3: Yes

Reviewer #1: Thank you for the opportunity to review this manuscript. The authors did an excellent job of addressing reviewer comments whenever possible and noting the rationale or limitations when they were not. Their efforts have strengthened an already-promising manuscript.

Reviewer #3: I have no additional comments. The paper is acceptable. The authors have addressed all the previous reviewers' comments.

**Do you want your identity to be public for this peer review?** For information about this choice, including consent withdrawal, please see our Privacy Policy

Reviewer #1: No

Reviewer #3: No

---

## [Editor Report · Acceptance letter]

PONE-D-24-50427R1

PLOS ONE

Dear Dr. Rowley,

I'm pleased to inform you that your manuscript has been deemed suitable for publication in PLOS ONE. Congratulations! Your manuscript is now being handed over to our production team.

Kind regards,

on behalf of

Dr. Mary Diane Clark

Academic Editor

PLOS ONE